# The Definition, Assessment, and Prevalence of (Human Assumed) Central Sensitisation in Patients with Chronic Low Back Pain: A Systematic Review

**DOI:** 10.3390/jcm10245931

**Published:** 2021-12-17

**Authors:** Ingrid Schuttert, Hans Timmerman, Kristian K. Petersen, Megan E. McPhee, Lars Arendt-Nielsen, Michiel F. Reneman, André P. Wolff

**Affiliations:** 1Pain Center, Department of Anaesthesiology, University Medical Center Groningen, University of Groningen, 9750 RA Groningen, The Netherlands; i.schuttert@umcg.nl (I.S.); h.timmerman02@umcg.nl (H.T.); 2Center for Neuroplasticity and Pain, Department of Health Science and Technology, Faculty of Medicine, Aalborg University, DK-9220 Aalborg, Denmark; kkp@hst.aau.dk (K.K.P.); mmp@hst.aau.dk (M.E.M.); lan@hst.aau.dk (L.A.-N.); 3Department of Medical Gastroenterology (Mech-Sense), Aalborg University Hospital, DK-9220 Aalborg, Denmark; 4Department of Rehabilitation Medicine, University Medical Center Groningen, University of Groningen, 9750 RA Groningen, The Netherlands; m.f.reneman@umcg.nl

**Keywords:** sensitisation, human assumed central sensitisation, HACS, nociplastic pain, quantitative sensory testing, QST, questionnaire, systematic review

## Abstract

Central sensitisation is assumed to be one of the underlying mechanisms for chronic low back pain. Because central sensitisation is not directly assessable in humans, the term ‘human assumed central sensitisation’ (HACS) is suggested. The objectives were to investigate what definitions for HACS have been used, to evaluate the methods to assess HACS, to assess the validity of those methods, and to estimate the prevalence of HACS. Database search resulted in 34 included studies. Forty different definition references were used to define HACS. This review uncovered twenty quantitative methods to assess HACS, including four questionnaires and sixteen quantitative sensory testing measures. The prevalence of HACS in patients with chronic low back pain was estimated in three studies. The current systematic review highlights that multiple definitions, assessment methods, and prevalence estimates are stated in the literature regarding HACS in patients with chronic low back pain. Most of the assessment methods of HACS are not validated but have been tested for reliability and repeatability. Given the lack of a gold standard to assess HACS, an initial grading system is proposed to standardize clinical and research assessments of HACS in patients with a chronic low back.

## 1. Introduction

The worldwide prevalence of chronic low back pain (CLBP) ranges between 2–25% [1,2,3]. According to the 2017 global burden of disease study low back pain is the leading cause of years lived with disability [4]. Recent evidence suggests that assessments of pain might represent underlying pain mechanisms and possibly help to identify patients at risk of poor response to different treatments for low back pain [5,6]. Currently, there are three mechanistic descriptors to describe patients’ pain: (a) nociceptive pain [7]; (b) neuropathic pain [7]; and (c) nociplastic pain [8]. Nociplastic pain has been suggested to cover a subset of patients with CLBP with widespread hyperalgesia, facilitated temporal summation of pain, and/or impaired conditioned pain modulation, without clear evidence of actual or threatened tissue damage, that are less likely to respond positively to standard pain treatments [9,10]. Before the term nociplastic pain was introduced, terms used included centralised pain [11,12], hyperresponsiveness [13], central hypersensitivity [14], and central sensitisation (CS) [11]. Additionally, multiple simultaneous mechanisms may play a role in patients with pain, which is also known as mixed pain [15]. In animal studies, various evolving definitions and descriptions of CS have been published previously. Examples include “the expression of an increase in excitability of neurons in the spinal cord” [16], and “central sensitisation amplifies and facilitates the synaptic transfer from the nociceptor central terminal to dorsal horn neurons” [17]. In 2011, CS was re-defined as: “an amplification of neural signalling within the central nervous system that elicits pain hypersensitivity” [18]. 

It has been suggested that there is an apparent conceptual overlap between CS and nociplastic pain, yet the terms stand for different entities [19]. CS refers to a neural mechanism, and nociplastic pain refers to a pain mechanism. However, both CS and nociplastic pain supposedly have altered nociception, which could originate either peripherally or centrally. The difference might be that in CS, altered nociception is mainly an increased responsiveness of the nociceptive neurons to their normal or subthreshold input in contrast with nociplastic pain in which either increased or decreased responsiveness is possible [20]. Studies with functional Magnetic Resonance Imaging (fMRI) support this by demonstrating hyperactive insula responses in most humans with syndrome/disorder/disease related to assumed CS [21,22]. 

In humans, the taxonomy published by the International Association for the Study of Pain (IASP) defined CS in 2008 as “Increased responsiveness of nociceptive neurons in the central nervous system to their normal or subthreshold afferent input” [7]. Recently, den Boer et al., 2019 defined CS as: “Hyperexcitability of the central nervous system” [23]. To date, the term CS lacks a structured definition based on expert consensus. Although the existence of CS is proven in animal studies, these assessments cannot be performed in humans. CS is proven in animals with direct electrophysiological recordings from central neurons which is not possible to perform in humans [24]. The principles found in animal studies have been applied to studies focusing on humans [24]. Therefore, CS should be regarded as a concept used to explain human pain conditions, adopted from animal studies. In order to address this and await proof of CS in humans, we have introduced the term Human Assumed Central Sensitisation (HACS) for studies in humans. 

Numerous clinical signs and symptoms may indicate HACS whereby differing assessment methods have been considered in the quantification of HACS in patients. Different proxies have been suggested for the assessment of HACS in patients, such as questionnaires [25], quantitative sensory testing (QST) [26], functional Magnetic Response Imaging (fMRI) [22], and brain-derived neurotrophic factor analysis (BDNF) [27,28]. QST, for example, is able to assess various phenomena that indicate altered pain processing such as secondary hyperalgesia, allodynia, impaired conditioned pain modulation (CPM), and facilitated temporal summation, which, based on preclinical evidence, could suggest alterations in central pain processing mechanisms [19]. 

A systematic review with an overview of clinimetric properties of used assessment methods is missing. To be able to evaluate the clinimetric properties of assessment methods suggested for assessing HACS, a gold standard must be established. Consequently, the prevalence and incidence of HACS in patients with CLBP seem to be scarcely reported. 

In neuropathic pain, for example, a grading system was created to address a lack of standardized and valid assessment methods and create a gold standard for the clinical assessment of neuropathic pain [29,30]. A comparable grading system can be developed for HACS.

This systematic review aimed (1) to investigate which definitions for HACS have been used, (2) to describe the methods that have been used to assess symptoms and signs for HACS in patients with CLBP, (3) to evaluate the clinimetric properties of the methods, and (4) to estimate the prevalence of HACS in patients with CLBP.

## 2. Materials and Methods

This systematic review was performed in accordance with the Preferred Reporting Items for Systematic Reviews and Meta-Analyses (PRISMA) Statement 2020 [31]. The protocol of this review was prospectively published on PROSPERO (CRD42019133226) Available online: https://www.crd.york.ac.uk/prospero/display_record.php?ID=CRD42019133226 (accessed on 23 October 2021).

### 2.1. Data Sources and Searches

A literature search was conducted on 19 February 2019 and updated on 7 January 2021, using the databases MEDLINE, EMBASE, PsycINFO, and CINAHL (see File S1). When articles use the term CS this will be referred to as HACS in this review. The search strategy included terms relating to HACS and CLBP. The terms were combined with the COSMIN MEDLINE filter for measurement properties of instruments [32]. The COSMIN filter for measurement properties was also translated to EMBASE [32], PsycINFO [32,33] and CINAHL [32]. For CINAHL and PsychINFO, there were two translations for the COSMIN filter, PsychINFO had one translation from COSMIN [32] and one translation from before COSMIN published their translation [34]. In CINAHL and PsychINFO, the search was performed twice, with each translation in a separate search. For the removal of duplicates when updating the search, the method described by Bramer et al. [35,36] was used.

### 2.2. Eligibility Criteria

Studies were included when an attempt was made to estimate HACS via subjective and objective markers, i.e., QST, questionnaires, or biomarkers. Included study designs were randomised controlled trials, cohort studies, case studies, cross-sectional and case-control studies. Moreover, any population described as CLBP or including patients with pain in the lumbar region lasting > 3 months were included. Patients with and without radiating pain as well as patients with specific and nonspecific CLBP were included. When there was a mixed population (CLBP and other pain syndromes) a study was included when the data of patients with CLBP could be extracted separately. Descriptions such as hyperresponsiveness or central hypersensitivity, which were assumed to reflect HACS, were also accepted. There was no restriction in the year of publication or language used. Animal studies, studies with children (below 18 years old), and studies with only healthy participants, were excluded. 

### 2.3. Study Screening

Two reviewers (IS & HT) screened the search results independently according to the eligibility criteria. Titles and abstracts were appraised, and when one or both reviewers were in doubt about the eligibility, the study was included. Full-text articles were retrieved through the university medical library or by requesting articles from the study authors. The same two reviewers appraised the eligibility of all full-text articles. References lists of the included articles were checked for potentially relevant articles. When there was disagreement about the study eligibility, a third reviewer (AW) provided the final judgement.

### 2.4. Risk of Bias Appraisal

The two reviewers independently appraised risk of bias based on the QUADAS-2 [37,38]. The QUADAS-2 consists of four key domains covering patient selection, index test, reference standard, and flow and timing. Each domain was assessed for risk of bias and applicability. Based on signalling questions (Appendix A), each item could be answered with “high risk of bias”, “low risk of bias”, or “risk of bias unclear”. In case of disagreement between the two reviewers, a third reviewer was asked for judgement.

### 2.5. Data Extraction

One reviewer (IS) performed all data extraction, which was cross-checked for accuracy by a second reviewer (HT). A standardised form was used to extract data from the included studies. Extracted data were divided into five categories: study descriptives and definitions, assessment methods, clinimetrics, and prevalence of HACS. Missing data were requested via e-mail from the study authors. When the study consisted of patients with various chronic pain condition(s), the data for patients with (solely) CLBP was requested by e-mail from the authors.

#### 2.5.1. Study Descriptives

The study descriptives included country, number of included patients, and patient characteristics (age, sex, and Body Mass Index (BMI)). The studies were divided into three groups: patients with CLBP only, patients with CLBP in combination with other pain condition(s) (CLBP+), and healthy volunteers (only when a comparison was made between patients with CLBP and healthy volunteers). 

#### 2.5.2. Definitions of Human Assumed Central Sensitisation

The definitions used for HACS and the source references for these definitions were collected.

#### 2.5.3. Assessment of Human Assumed Central Sensitisation

Methods for the assessment of HACS were included when they were used for the assessment of HACS in the methods section of the included study. Studies that proposed new assessment methods or a translation of a current assessment method for HACS were also eligible for appraisal if this was stated in the introduction. Potential new assessment methods for HACS mentioned in the discussion section of the studies were excluded because the initial use, as described in the method section, of the assessment method was different in the study. Therefore the assessment method was not examined or validated as an assessment method for HACS in that study.

#### 2.5.4. Clinimetrics of Human Assumed Central Sensitisation Assessment Methods

Clinimetrics outcomes, including sensitivity, specificity, accuracy, internal consistency, the area under the curve, likelihood ratio, predictive values, and test-retest reliability, were retrieved from the included studies. Associations with other assessment methods in these studies, independently if these other assessment methods were HACS-related, were extracted. In this review, a correlation coefficient of less than 0.29 was deemed low, from 0.3 to 0.49 was considered moderate, from 0.5 to 0.69 substantial, from 0.70 to 0.99 very high, and 1.0 perfect [39,40]. 

#### 2.5.5. Prevalence of Human Assumed Central Sensitisation

The prevalence of HACS was retrieved when explicitly reported in eligible studies. Additionally, the CSI part A scores were extracted to assess the prevalence of HACS using a cut-off score of 40, according to previous suggestions [25].

### 2.6. Data Synthesis

All extracted data were synthesised to provide an overview of the information for each study aim, study characteristics, and heterogeneity in the study findings. To calculate the prevalence of HACS in patients with CLBP, the CSI scores, presented in the articles or provided by the authors, were used. Both the calculated prevalence and those presented in the included articles were also used to calculate an overall prevalence. When available, the prevalence for CLBP only and CLBP in combination with other pain condition(s) were calculated separately. 

## 3. Results

A total of 12,764 articles were identified. Checking the reference lists of the included studies resulted in 280 more articles. After the removal of duplicates, 8772 articles remained, and 507 titles were selected to be screened on abstract. 153 articles were included in the full-text screening. This resulted in the inclusion of 34 studies in this review, as shown in the PRISMA Flow Diagram (Figure 1).

### 3.1. Study Characteristics 

Table 1 shows descriptive data from the included studies. These data represent the number of patients with CLBP only, CLBP+, and healthy controls. In 16 out of 34 studies, it was not specified whether patients either had CLBP only or CLBP+, in these studies ‘CLBP’ was stated because no distinction could be made from the data provided. For all groups, the number of patients, age, sex, and BMI (when applicable) are presented (Table 1).

### 3.2. Risk of Bias

The results of the QUADAS-2 are shown in Table 2 and Figure 2. Of the 34 included studies, 33 did not describe a reference standard for HACS. Therefore, this was assessed as a high risk of bias for risk of bias and applicability. For the risk of bias, it was not possible to assess “flow and timing” [25,41,42,43,44,45,46,47,48,49,50,51,52,53,54,55,56,57,58,59,60,61,62,63,64,65,66,67,68,69,70,71,72,73]. Only 1 study used a reference standard, however, this study did not compare its results to an index test for HACS [71]. The reference standard they used was a clinical judgement which, as they noted, may be used as an appropriate alternative reference standard in the absence of a ‘diagnostic’ gold standard [71,74,75]. An index test was not compared to a reference standard for HACS in any of the included studies, hence “flow and timing” was noted as not applicable.

### 3.3. Definition of Human Assumed Central Sensitisation

In the studies included in this review, there were various definitions postulated for HACS. In 3 of the studies, no definition was provided [61,72,73]. The references used for the definitions were retrieved to find the similarities and differences between the different definitions. In total, 40 definition references were used to define HACS. The number of definition references used per study varied between 1 and 9. Most definitions referenced that used by Woolf (2011) [18]: “an amplification of neural signalling within the central nervous system that elicits pain hypersensitivity” (*n* = 17). The second most frequently used definition reference was from IASP (2008) [7]: “Increased responsiveness of nociceptive neurons in the central nervous system to their normal or subthreshold afferent input” (*n* = 6). The third most frequently used definition reference was from Nijs et al. (2010) [76]: ”an augmentation of responsiveness of central neurons to input from unimodal and polymodal receptors“ (*n* = 5). All other definition references were used three times or less, as shown in Table 3.

### 3.4. Assessment of Human Assumed Central Sensitisation

Four questionnaires and 16 QST methods to assess HACS were observed (Table 4). In Table 5, correlations greater than or equal to 0.5 between HACS assessment outcome and other assessment methods from the studies, are shown (Table 5). Some assessment methods have described correlations greater than or equal to 0.5 as well as below 0.5 in other studies. This counts for the Oswestry Disability Index, the physical and mental component of the short form-36, NRS pain intensity, McGill Pain Questionnaire, Pain Catastrophizing Scale, sensory profiles: sensory seeking and low registration, and State-Trait Anxiety Inventory. All correlations are presented in Appendix A.

#### 3.4.1. Questionnaires

The CSI [25] was used to assess HACS in 22 of the 34 studies (64.7%) [25,41,42,46,47,48,49,50,55,56,57,58,59,60,61,62,64,65,67,68,69,70]. The CSI was developed to identify key symptoms associated with syndrome/disorder/disease related to HACS and quantifies the degree of these symptoms [25]. Cross-cultural adaptations of the CSI included in this review were performed in Dutch [59], Greek [47], Italian [48], Nepali [70], and Serbian [57]. When assessing the clinimetric values of the CSI, three studies [48,57,59] did not compare the CSI results to a different assessment method. 

The Michigan Body Map (MBM) [107] was used in one study [62] to assess HACS. The MBM is a patient-reported outcome measure to assess body areas where chronic pain is experienced and to quantify the degree of widespread body pain when assessing pain features. The MBM, was used to determine the extent to which pain had spread throughout the body as a possible result of HACS [62]. If only one body area was affected, the likelihood that HACS was present was minimal, but the more body areas affected, the greater the chance that HACS was present.

The short-form McGill Pain Questionnaire revised version (SF-MPQ-2) [109] was used for the assessment of HACS in one study [62]. The SF-MPQ-2) has a mechanism-based approach to assess neuropathic pain [108,109], and was used in this study as a proxy for HACS [62]. The authors, however, did not specify how the SF-MPQ-2 was used to assess HACS. 

The Widespread Pain Index (WPI) [110,112] was used in one study to determine the extent that pain had spread throughout the body as a possible result of HACS [43]. With an increasing number of affected body areas, the possibility of the presence of HACS also increases. In this study, the 2011 FM survey, a diagnostic tool for FM, consisted of both the WPI and symptom severity [110,111]. 

#### 3.4.2. QST Measures

Pressure pain threshold (PPT) was assessed using a handheld algometer at up to four different anatomical locations [43,44,60,61,64,72,73]. The PPT was conducted using a push force gauge whereby pressure applied over a particular muscle or tissue is increased until the patient describes the pressure as becoming painful. In these studies, HACS was indicated when PPTs were lower in patients with CLBP compared to healthy controls. PPT was assessed in the originally affected area or on distal test areas, for example, the opposite side of the body or another body part. The anatomical assessment locations differed by study, and the number of locations assessed differed with a range of 1 to 4 locations (Table 4). In three studies, patients with CLBP were compared with healthy volunteers [43,72,73]. 

Temporal summation (TS) or wind-up ratio was used four times [54,64,72,73] to estimate HACS. As wind-up ratio and TS refer to the same phenomenon, the term TS is in this review used for both. Three different methods were used to assess TS: (1) TS with heat pain [54], (2) TS with predetermined PPT [64], and (3) TS with pinprick [72,73]. To calculate TS, the mean pain ratings of the series were divided by the mean pain ratings of single stimuli. TS was considered positive when the pain intensity increased within the test series. 

Conditioned pain modulation (CPM) [43,60,61] was used in three out of thirty-four studies to assess HACS. CPM is a human proxy assessment to evaluate the balance between the descending inhibitory and facilitatory pathway activity [43,60,61,113,114]. For CPM, the effect of applying a second painful stimulus in addition to the first was assessed; the first painful stimulus is termed the test stimulus, and the second painful stimulus was referred to as the conditioning stimulus. A comparison between test stimuli can be made before, after [60], and during the conditioning stimulus [43,60]. Two different conditioning stimuli were utilized by the studies in this review. These were: (1) a pressure cuff [43], and (2) ice water [60,61]. PPT was used in all the studies as a test stimulus in different locations; thumbnail [43], lower back [43], upper leg [60], and 2nd toe [61].

Thermal QST [54,72,73] was used in three studies to assess HACS. For thermal QST a thermal sensory testing device was used [54,72,73] to assess cold detection threshold [72,73], cold pain threshold [54,72,73], cold pain tolerance [54], warmth detection threshold [72,73], heat pain threshold [54,72,73], heat pain tolerance [54], thermal sensory limen [72,73], and paradoxical heat sensations [72,73]. To indicate the presence of HACS in patients with CLBP, the threshold must be lower on the affected areas and at the distal test areas, relative to control subjects.

Other QST measurements. In two studies [72,73], in addition to the previously mentioned QST measures, extra QST measures were obtained (Table 4), including (A) mechanical detection threshold, (B) Mechanical pain threshold, (C) Mechanical pain sensitivity, and dynamic mechanical allodynia, and (D) Vibration detection threshold. According to these studies, the QST measures assess all relevant aspects of the somatosensory system, including large and small fibre functions and signs of central sensitisation.

Questionnaires and QST. In five studies [43,44,60,61,64], both questionnaires and QST measures were used to assess HACS. 

Although authors claimed to address HACS in six studies [45,52,53,63,66,71], they did not describe assessment methods that could estimate HACS in the method section of their study. 

### 3.5. Clinimetrics of Human Assumed Central Sensitisation Methods

7 studies (20.6%) [25,47,48,57,59,64,70] reported on the clinimetrics of methods used to assess HACS. 6 studies reported an internal consistency of 0.87 or higher [25,47,48,57,59,70] and 5 studies reported a test-retest reliability of 0.82 or higher [25,47,57,59,70]. Sensitivity and specificity for the assessment of HACS and syndrome/disorder/disease related to HACS were calculated in one study [64]. HACS was assessed based on the CSI part A with a sensitivity of 73.7% and specificity of 37.7% when using a cut-off score of 20 (0–100) [64]. Syndrome/disorder/disease related to HACS assessed were based on the CSI part B. A cut-off score of 28 (0–100), resulted in a sensitivity of 69.2% and a specificity of 69.2% [64] to assess HACS in patients with CLBP (Table 4).

### 3.6. Estimation of the Prevalence of Human Assumed Central Sensitisation

The prevalence of HACS in patients with CLBP was described in three studies, as shown in Table 3 (Table 3). The CSI scores resulted in a prevalence of 71.1% (*n* = 128) [46]. The presence of tactile allodynia resulted in a prevalence of 60.8% (*n* = 74) in patients with CLBP+, a prevalence of 13.3% (*n* = 15) in patients with CLBP only [52] and PPT combined with TS resulted in a prevalence of 18.3% (*n* = 104)) [64]. Based on these 3 studies (*n* = 321), the mean prevalence of HACS in patients with CLBP was 48.9%. 

Additionally, calculations were based on CSI scores with a cut-off value of 40 (0–100) in 16 studies [41,46,47,49,50,51,55,56,57,58,59,65,67,68,69,70] with a total of 2347 patients with CLBP, the prevalence for HACS in patients with CLBP was 43.2%. In 8 studies [47,51,56,58,59,67,68,70], the prevalence of HACS, based on CSI scores with a cut-off value of 40, could be compared between a group with CLBP only and the CLBP+ group. In our review, we calculated that the prevalence of HACS based on the CSI was not different between the group with CLBP+ (prevalence of 41.9%, 343 patients out of 819 included patients with CLBP) and the group with CLBP only (prevalence of 41.2%, 289 patients, out of 701 included patients with CLBP).

## 4. Discussion

The aim of this systematic review was to highlight outcome assessments used for HACS in patients with CLBP. As a gold standard for HACS assessment does not exist, it was important to gauge the various terminologies applied to HACS as well as assessment methods in humans used in literature. A variety of definitions and assessment methods were observed. Evidence supporting the preference for a particular assessment method or a combination of assessment methods was not found. The QUADAS-2 showed for risk of bias mostly unclear or low risk of bias scores. The reference standard was scored in almost all studies as a high risk of bias because only once a reference standard was used. There was no comparison between the index test and the reference standard, therefore all flow and timing were not applicable. Applicability concerns mainly showed a low risk of bias for patient selection and index test, and the reference standard was mostly a high risk of bias, because no reference standard was used.

### 4.1. Definition of Human Assumed Central Sensitisation

Across all the studies concerning HACS, a wide variety of definitions for CS are described. At this moment it can’t be said which definition of CS in patients is to be used. Most definitions as described in the included studies in this review are more explanatory rather than really defining central sensitisation with all the associated issues. This is because central sensitisation is more than the outcome of just one assessment. Moreover, central sensitisation might also reflect an adaptive primary mechanism or may occur as a secondary response following an (acute) injury [80]. Finally, (nociplastic) pain is not the same as central sensitisation and central sensitisation might not be the main cause of chronic pain [115].

The definition proposed by the International Association for the Study of Pain [100] cannot directly assessable in humans, as direct recordings from central neurons based on electrophysiological methods are not possible yet. Therefore the term central sensitisation should not be used as such in both human research and clinical practice [24]. 

Although an HACS-related term was used in many articles included in this review, these terms were only stated in the introduction and discussion sections of six articles [45,47,51,56,63,66]. This gives the impression that the authors used HACS as an explanatory model for the findings from their assessment methods, but lacked a clear explanation of how they considered and defined HACS. Twenty studies, did not use an HACS-related term in the results section [25,42,45,47,48,50,51,53,54,56,58,60,61,63,65,66,67,68,72,73], and therefore did not provide additional value for the assessment of HACS, the clinimetrics of assessment methods and/or the prevalence of HACS. The variety of definitions and ambiguous terminology used in the literature made the assessment of HACS even more unclear because a lack of unequivocal definitions and terms makes the assessment of HACS more confusing and incorrect. In 2019 a more comprehensive definition for CS was presented: “Hyperexcitability of the central nervous system” [23]. This definition assumed that the central, not peripheral, nervous system is the main contributor to HACS. However, this cannot be established in most cases of HACS, because altered sensation can be caused peripherally as well as centrally. It is therefore clear that a unified definition for HACS was not utilized by the included studies. 

### 4.2. Methods for Assessing Human Assumed Central Sensitisation

The manifestation of HACS might refer to the possible presence of CS as a neural mechanism in humans. HACS can present itself in many symptoms and/or manifestations such as altered CPM, decreased pain thresholds, increased temporal summation, and the presence of widespread pain) but can, until today, not directly be demonstrated in humans [24]. In three studies, patients were appointed to groups with and without HACS based on CSI or QST assessment outcomes [46,52,64]. In all other studies, methods were used to assess HACS, but no conclusion about the presence or absence of HACS could be drawn. The CSI is a questionnaire that is often used, also in the included studies of this review, and can be a quick tool for identifying whether symptoms may be associated with HACS or syndrome/disorder/disease related to HACS. High CSI scores may suggest HACS or suggest the need to perform additional tests to assess syndrome/disorder/disease related to HACS, after which appropriate treatment can be initiated [116]. However, because of the lack of a reference standard to date, no scientific evidence to support the value of the CSI as an indicator for HACS can be found [64,117]. Hence, the interpretation of the CSI data should be made cautiously. None of the included studies provided evidence, by relating the CSI to other HACS assessments, that could establish a significant role for the CSI in the assessment of HACS. Moreover, the cut-off score is established at 40 out of 100 [84]. It should be noted that there is a lack of cut-off scores for QST resulting from highly variable assessment methods between research groups and a lack of high-quality, large sample normative data. Different assessment methods can be compared to each other, such as questionnaires with physical assessment methods. In this review, only one significant substantial negative/inverse correlation was found (CSI and lifting capacity) [42]. With respect to the various methods assessing HACS, this review did not find clear associations between assessment methods for HACS or other non-HACS assessments methods in the studies. It should be noted that some studies discuss the presence of HACS based on assessment methods used in the study without initially describing this in the method section. These were excluded from this analysis because validation or examination of these assessments for HACS was not possible.

### 4.3. Clinimetrics of Human Assumed Central Sensitisation Assessment Methods

Clinimetrics concerning only reproducibility and consistency were calculated in 7 studies. Sensitivity and specificity were reported only once [64]. The clinimetric outcomes reported in most included studies concerned internal consistency and test-retest reliability which were high for the CSI. Sensitivity and specificity of the assessment methods are required to address the validity of an assessment method; in the included studies they were, however, not described likely due to the aforementioned lack of diagnostic gold standard.

A new cut-off value of 20 (0–100) for the CSI for patients with HACS was presented in the study of Mibu et al. [64] with a sensitivity of 73.7% and a specificity of 37.7%, not supporting this new cut-off as being sufficient to discriminate for HACS. Neblett et al. [84] reported a sensitivity of 81% and a specificity of 75% for the CSI, but included patients with chronic pain and not only patients with CLBP. In the same study, a non-patient group (undergraduate students not being treated for chronic pain) was compared with a group of patients with one or more syndrome/disorder/disease related to HACS in order to identify the cut-off value [84]. Nebletts’ study does not improve the potential discriminability of HACS with the CSI, as it was not tested in a group of patients with chronic pain only. Mibu et al. [64] based the presence of HACS on the PPT tertile data and a positive TS value. Using the lowest tertile of the PPT score can give a distorted picture because the cut-off value may change and vary based on the assessed population. 

### 4.4. The Estimated Prevalence of Human Assumed Central Sensitisation

The prevalence estimates 43.2% of HACS in patients with CLBP in the literature [41,42,46,47,49,50,51,55,56,57,58,59,65,67,68,69,70] should be interpreted as being based on the assumption that specific symptoms and signs reflect mechanisms that are observed in animal experiments and could explain pain symptoms for a subset of patients experiencing CLBP. Therefore, we introduced the term HACS and have estimated its prevalence in patients with CLBP as derived from proxy assessments. 

Few studies in this review estimated the prevalence of HACS, which varied between 13% and 78% [46,52,64]. These prevalence estimates should be interpreted with caution as calculations were based on several different assessment methods, i.e., CSI scores, tactile allodynia, and PPT scores combined with TS scores. The CSI is so far the only assessment method with an established cut-off value, however, the CSI has been suggested to show a stronger association to pain severity and psychopathology than with QST measures of HACS [118]. All studies are inherently limited due to a lack of a gold standard and, thus, limited available measurement properties. Therefore, it should be noted that the discriminability and, consequently, the prevalences are debatable.

To obtain a better understanding of the effects of co-existent chronic pain condition(s) on the prevalence of HACS in patients with CLBP, the prevalence of HACS between patients with CLBP only and patients with CLBP in combination with other pain condition(s) was compared. These prevalence estimates were based mostly on the CSI, however, the prevalence based on tactile allodynia showed similar scores. Due to the low number of studies (*n* = 8) and the low number of patients (CLBP only *n* = 708 and CLBP + *n* = 752) used to estimate prevalence in this review, these findings should be interpreted with caution. 

### 4.5. Limitations

This review focused on patients with CLBP, but in some studies, patients were also included with CLBP in combination with other pain condition(s). Due to the complex nature of chronic pain, and often difficulties in diagnosing it, pain condition(s) were not always well-documented in the included studies. It is possible that HACS is presented more in patients with neuropathic pain, this might result in less generalisability of the data. But the presence of neuropathic pain is not well documented in all included studies, therefore, a separate analysis was not possible. Therefore, forming the analysis groups for studies with ‘CLBP only’ or “CLBP in combination other pain condition(s)” could only be completed in 16 studies [41,42,45,46,49,50,53,54,60,61,64,65,66,69,72,73]. The possible presence of multiple pain conditions in patients may have influenced the results. Similarly, HACS was considered as a static phenomenon with prevalence estimates based on a single assessment. Although reliability was high for HACS assessment methods when reported, recent work has suggested that psychological symptoms associated with the CSI [105], as well as PPT and TS [119], may fluctuate with pain severity. For now, it is still unclear if HACS is a state or a trait. Both human and animal studies would suggest the presence of CS changes over time depending on pain presence and pain progression [19].

It was part of the original setup of this study to do a meta-analysis. However, because of the vast number of varying assessments methods found in all included studies, while each of these methods was applied just once or a few times it was impossible to do this analysis.

The risk of bias assessment based on the QUADAS-2 was challenging to use in combination with the selected studies. The lack of a gold standard for defining and estimating HACS, the reference standard, and flow and timing domains hinders adequate judgement. However, this was the case for all studies and did not affect the comparison of the risk of bias assessment between papers. The Newcastle–Ottawa–Scale [120] was considered as a possibility to be used in the assessment of the risk of bias, but none of the studies were cohort or case-control studies. No other suitable risk-of-bias assessment method for diagnostic accuracy studies was found.

### 4.6. Gold Standard 

To date, CS cannot according to the IASP taxonomy be demonstrated in humans, and HACS as a proxy lacks a uniform definition and is an unambiguous way of assessment. The lack of a gold standard in patients with CLBP has led to the application of a large variety of methods to assess HACS as shown in this review. Lack of clarity on how to assess HACS is illustrated by the finding that QST is referred to as a gold standard in one of the studies [121]. However, in other studies this statement is refuted [48,55,59,60,68,71,80,122,123]. Until CS can be demonstrated in humans a gold standard cannot be established. Researchers are still searching for an acceptable proxy to establish HACS [122]; examples are the CSI [48,55,59,60,123], exclusion of neuropathic pain and the differential classification of nociceptive versus HACS pain [81,124], QST measures [60,68] and clinical judgement [64]. For patients with CLBP, one study was included in this review where the assessment methods were compared with an expert’s opinion [71]. In this study, based on an expert’s opinion, patients with CLBP were classified founded on posture, movement, and neurological assessment, and a clinical interview. Here the authors did an attempt to assess HACS by excluding another mechanistic pain that did not apply. 

Another major challenge and limitation is to differentiate HACS from peripheral sensitisation. When comparing assessments on various body locations, for example, distant from the primary pain location, it should be noted that an assessment such as PPT might reflect peripheral sensitisation when assessed in only one or two locations and more assessment locations are recommended. As shown in this review, there are many studies with different definitions and different assessment methods, leading to inconclusive results, which shows a need for a uniform way of assessing HACS. The question arises if and how the expert’s opinion can be used as a reliable assessment to be considered a proxy to assess HACS. To meaningfully contribute, this expert opinion should be based on a uniform definition for HACS, a uniform way of HACS presentation, and a uniform way of assessing HACS. 

### 4.7. A Grading System for Human Assumed Central Sensitisation

The scientific discussion about the assessment of neuropathic pain is much more advanced than for the assessment of HACS. In neuropathic pain, a diagnostic gold standard for assessment is also lacking, and although many questions are still unanswered, there seems to be some more understanding of this clinical condition compared to HACS. A grading system provides a more uniform way to grade the probability of neuropathic pain in patients in daily clinical practice as well as in research. This grading system for neuropathic pain, firstly described by Treede et al. [30] and updated by Finnerup et al. [29], is widely accepted and serves as a basis to propose an idea to assess HACS more uniformly: a grading system for HACS (Figure 3). 

Similar to the neuropathic pain grading system, the idea behind a grading system for HACS is to incorporate the various definitions of HACS, the available (clinical) tests, and the (validated) assessment methods to become a more directive assessment of HACS in daily clinical practice as well as in (clinical) research. The grading system for HACS as presented here should be further discussed, elaborated on, further developed, and validated for clinical and scientific use in future studies. 

At the first level, patients with CLBP are graded based on the patient’s history. When patients’ pain is only partly explained by anatomical, morphological and neurological lesions, and/or diseases or fully unexplained, there is a possibility of HACS. This is because HACS can be present in patients with nociplastic pain, but also in patients with nociceptive or neuropathic pain. If this criterion is fulfilled, the first level of certainty—“possible” HACS is reached. 

The second level is based on the indicator tests (see Table 6). With an increasing number of positive indicator tests, as described in this review, the description of “probable” HACS may be relevant. When one or no indicator tests are fulfilled, the patient has “unconfirmed” HACS. Among the described tests to assess HACS in this review, there are assessments that can be used as an indicator test for HACS in patients with CLBP. Candidates are the CSI, PPT, CPM, and TS. It should be noted that there is currently only a cut-off value for the CSI established based on previously performed validation studies. For the other indicator tests, the validity should be established to be able to distinguish between the presence or absence of HACS. Future studies should also be aimed at assessing the validity of multi-modal test batteries to assess HACS. 

The third level should grade patients with CLBP based on diagnostic tests or confirmatory tests as “definite” HACS. To assess the level of “definite” HACS, more scientific research is required because there are currently no diagnostic methods to assess HACS which are validated against a gold standard. This third level might be added in the future when confirmative assessment methods are available. However, promising insights with respect to HACS are already shown in brain imaging such as fMRI [22] or blood serums such as BDMR [27,28].

### 4.8. Limitations Initial Grading System

HACS cannot and should not be seen as a ‘diagnosis’ in humans. Assessment of HACS via the grading system will raise the possibility of the presence of HACS which is based on the different manifestations and symptoms related to HACS. In addition, raising the possibility of HACS is not based on a single, specific, assessment but is based on following a process using various assessment methods. Future research is needed to further validate, improve, and expand the initial grading system for HACS. This is to create a uniform process in the assessment of HACS. This will lead to a more comparable (clinical) assessment of patients, which will benefit daily clinical practice as well as research on this topic. As an example of such studies, previously performed forward and backward translations between animal studies and human studies regarding the assessment of CS showed that the cuff-algometer test for the assessment of CPM is similar to what is assessed in animals (rats) and in humans [125]. For the suggested indicator tests (PPT, WUR, CPM) clinically relevant values should be established. Other assessment methods that are less frequently used to assess HACS should be evaluated for possible expansion of the indicator tests. Furthermore, confirmatory tests should be studied to confirm the presence of HACS. With the addition of confirmatory tests, the indicator tests could be improved. However, it is still needed to find objective assessment methods for HACS and a gold standard method that can be used in daily clinical practice. 

## 5. Conclusions

In this systematic review, various definitions of HACS and assessment methods are discussed with a specific focus on CLBP. It may be concluded that the definitions of HACS that apply to chronic pain in humans, in general, show no uniformity in consensus. Moreover, there is no standardised way to assess HACS in patients with CLBP based on current assessments that use various proxies. Given the fact that CS cannot, according to the definition, be measured directly in humans, further research is needed to develop reliable proxies. A gold standard to diagnose HACS is lacking, undermining the validity of the actually available indicator tests for HACS in patients with CLBP, but an initial grading system for HACS is proposed. 

## Figures and Tables

**Figure 1 jcm-10-05931-f001:**
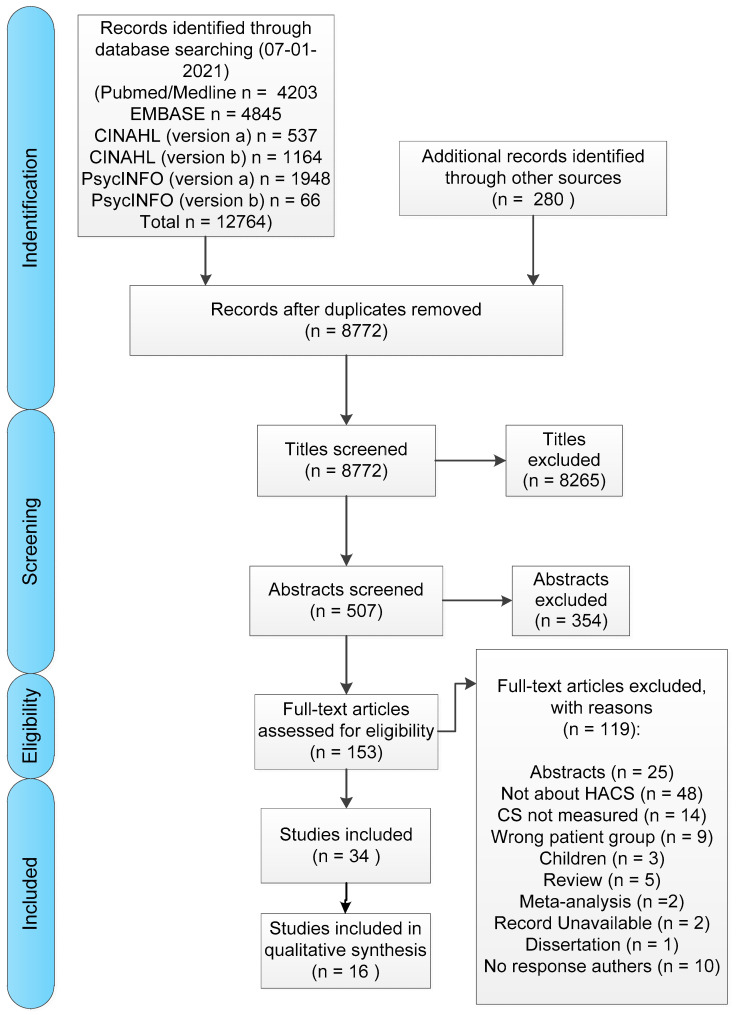
PRISMA Flow Diagram. CS: Central Sensitisation, HACS: Human Assumed Central Sensitisation.

**Figure 2 jcm-10-05931-f002:**
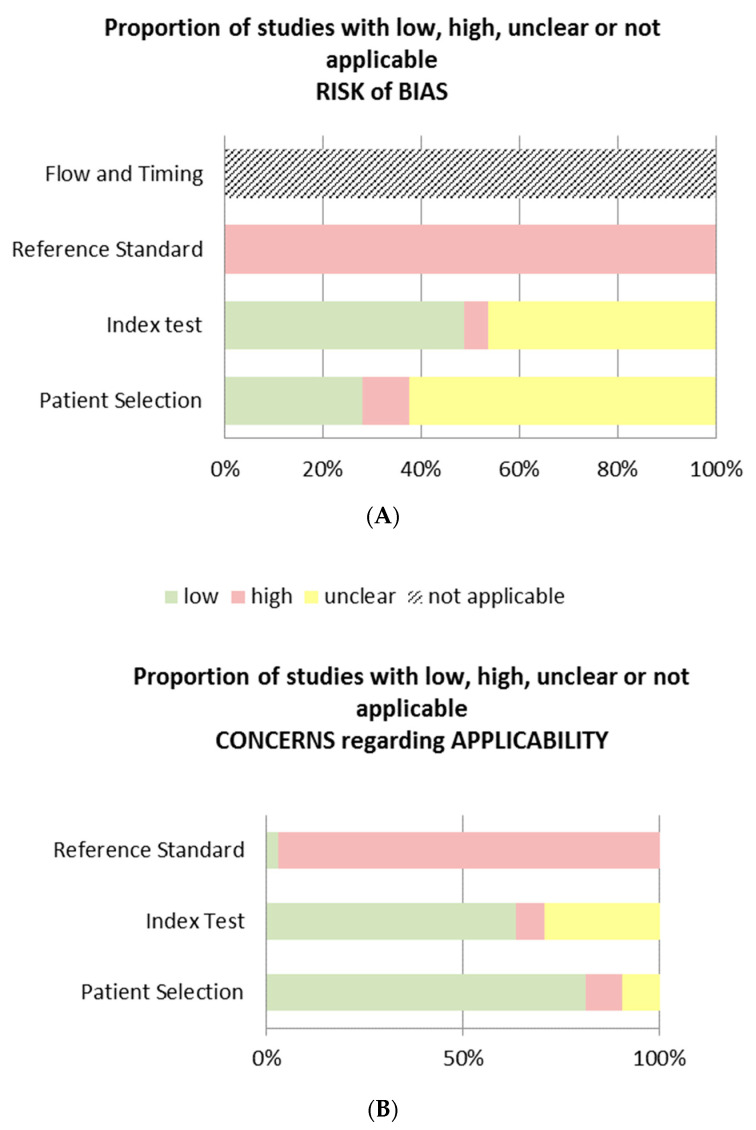
Proportions of studies with low risk of bias, high risk of bias, unclear or not applicable. (**A**): Risk of bias; (**B**): Concerns regarding applicability.

**Figure 3 jcm-10-05931-f003:**
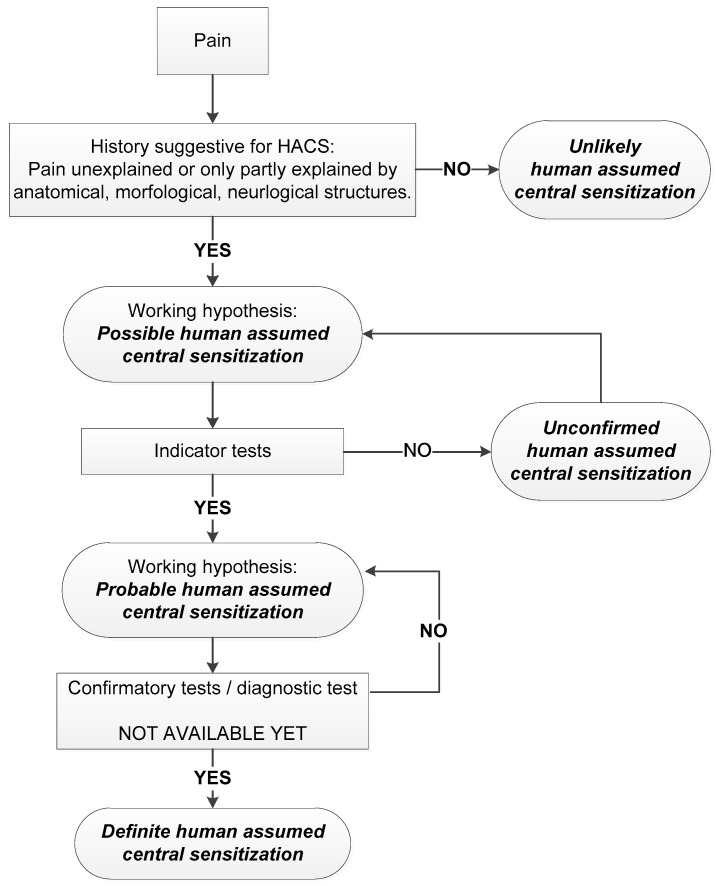
Grading system for human assumed central sensitisation for patients with chronic low back pain.

**Table 1 jcm-10-05931-t001:** Descriptives of the included studies.

1st Author, Year		Country	nr. of Participants	Age (Mean ± SD/Range/95% CI)	Sex (%Female)	BMI (Mean ± SD/95% CI)
Ansuategui Echeita, 2020a *^,^#	[41]	The Netherlands	CLBP: 56	CLBP: 42.55 ± 13.22	CLBP: 33 (58.9%)	CLBP: 26.30 ± 4.77
Ansuategui Echeita, 2020b *^,^#	[42]	The Netherlands	CLBP: 56	CLBP: 42.55 ± 13.22	CLBP: 33 (58.9%)	CLBP: 26.30 ± 4.77
Aoyagi, 2019	[43]	United States of America	CLBP only: 24 CLBP+: 22 Healthy controls: 22	CLBP only: 42.38 ± 12.37 CLBP+: 43.95 ± 14.00 Healthy controls: 41.15 ± 8.83	CLBP only: 15 (63%) CLBP+: 17 (77%) Healthy controls: 15 (68%)	CLBP only: 28.76 ± 6.20 CLBP+: 31.34 ± 6.13 Healthy controls: 28.35 ± 8.10
Aoyagi, 2020	[44]	United States of America	CLBP only: 30 CLBP+: 30	CLBP only: 42.38 ± 12.37 CLBP+: 41.23 ± 13.81	CLBP only: 19 (63%) CLBP+: 24 (80%)	CLBP only: 29.58 ± 6.43 CLBP+: 31.18 ± 6.67
Ashina, 2018, #	[45]	Denmark	CLBP: 570	CLBP: 48.31 ± 0.57	CLBP: 305 (53.5%)	NR
Bid, 2017	[46]	India	CLBP: 128 Experimental Group (*n* = 64) Control Group (*n* = 64)	CLBP: Experimental Group: 41.33 ± 7.27 CLBP: Control Group: 41.12 ± 7.76	CLBP: Experimental Group: 36 (56.25%) CLBP: Control Group: 42 (65.63%)	CLBP: Experimental Group: 24.88 ± 2.97 CLBP: Control Group: 24.72 ± 2.76
Bilika, 2020, #	[47]	Greece	CLBP only: 28 Healthy controls: 50	CLBP only: 49.04 ± 14.811 Healthy controls: 27.90 ± 8.707	CLBP only: 17 (60.7%) Healthy controls: 25 (50%)	NR
Chiarotto 2018, #	[48]	Italy	CLBP only: 76	CLBP only: 50.9 ± 13.7	CLBP only: 56 (73.7%)	CLBP only: 24.68 ± 4.30
Clark, 2018	[49]	New Zealand and United Kingdom	CLBP: 21	CLBP: 43 (range 20–64)	CLBP: 16 (76%)	NR
Clark, 2019, #	[50]	United Kingdom, Ireland and New Zealand	CLBP: 165	CLBP: 45 ± 12	CLBP: 126 (76%)	NR
Cuesta-Vargas, 2016, #	[51]	Spain	CLBP only: 126 CLBP+: 90	CLBP only: 52.50 ± 12.61 (10 missing) CLBP+: 57.50 ± 12.28 (6 missing)	CLBP only: 14 (11.1%) (84 missing) CLBP+: 17 (18.9%) (59 missing)	CLBP only: 25.70 ± 4.23 (8 missing) CLBP+: 26.02 ± 3.89 (3 missing)
Defrin, 2014	[52]	Israel	CLBP only: 15 CLBP+: 74 Healthy controls: 22	CLBP only: Axial CLBP: 64.5 ± 20.7 CLBP+: CLBP with radiation: 65.8 ± 12.9 Healthy controls: 54.2 ± 18.6	CLBP only: Axial CLBP: 6 (40%) CLBP+: CLBP with radiation: 39 (53%) Healthy controls: 12 (55%)	NR
Dixon, 2016	[53]	United States of America	CLBP: 59 Healthy controls: 44	CLBP: 40.56 ± 11.32 Healthy controls: 40.26 ± 11.6	CLBP: 27(46%) (4 missing) Healthy controls: 24(55%) (2 missing)	NR
Hubscher, 2014	[54]	Australia	CLBP: 30 Healthy controls: 30	CLBP: 30.6 (range 21.8–35.0) Healthy controls: 28.0 (range 21.8–31.0)	CLBP: 15 (50%) Healthy controls: 17(56.7%)	NR
Huysmans, 2018	[55]	Belgium	CLBP only: 38	CLBP only: 40.76 ± 13.30	CLBP only: 24 (63.2%)	CLBP only: 24.98 ± 3.16
Ide, 2020, #	[56]	Japan	CLBP only: 46 CLBP+: 206	CLBP only: 74.33 ± 7.57 CLBP+: 75.95 ± 7.67	CLBP only: 24 (52.2%) CLBP+: 140 (68.0%)	CLBP only: 22.96 ± 2.74 CLBP+: 22.62 ± 3.16
Knezevic, 2018, #	[57]	Serbia	CLBP only: 157 CLBP+: 74	CLBP only: 51.59 ± 13.34 CLBP+: 56.65 ± 9.55	CLBP only: 89 (56.7%) CLBP+: 57(77%)	NR
Knezevic, 2020, #	[58]	Serbia	CLBP only: 155 CLBP+: 88 Healthy controls: 146	CLBP only: 51.74 ± 13.44 CLBP+: 56.77 ± 9.49 Healthy controls: 39.18 ± 14.95	CLBP only: 83 (53.5%) CLBP+: 66 (75.0%) Healthy controls: 102 (69.9%)	NR
Kregel, 2016, #	[59]	The Netherlands and Belgium	CLBP only: 4 CLBP+: 11	CLBP only: 51.50 ± 15.97 CLBP+: 40.45 ± 9.20	CLBP only: 3 (75.0%) CLBP+: 8 (72.7%)	NR
Kregel, 2018	[60]	Belgium	CLBP: 54	CLBP: 41.24 ± 13.04	CLBP: 31 (57.4%)	NR
Leemans, 2020	[61]	Belgium	CLBP: 50 Experimental (*n* = 25) Control (*n* = 25)	CLBP: Experimental: 43.9 ± 12.2 CLBP: Control: 44.7 ± 12.2	CLBP: Experimental: 13 (52%) CLBP: Control: 14 (56%)	CLBP: Experimental: 26.5 ± 3.8 CLBP: Control: 27.6 ± 5.1
Mayer, 2012	[25]	United States of America	CLBP only: 44 Healthy controls: 40	CLBP only: 42.8 ± 10.0 Healthy controls: 21.33 ± 13.6	CLBP only: 11 (25%) Healthy controls: 31 (77%)	NR
McKernan, 2019, #	[62]	United States of America	CLBP only: 38	CLBP only: 46.75 ± 13.74	CLBP only: 24 (63.2%) (2 missng)	NR
Mehta, 2017	[63]	United Kingdom	CLBP+: 23 Healthy controls: 21	CLBP+: 46 Healthy controls: 60 (range 40–81)	CLBP+: 13 (56.5%) Healthy controls: 17 (81.0%)	NR
Mibu, 2019	[64]	Japan	CLBP: 104	CLBP: 58.4 ± 14.2	CLBP: 77 (74.0%)	NR
Miki, 2020	[65]	Japan	CLBP: 238	CLBP: 63.50 ± 16.0	CLBP: 102 (42.9%)	CLBP: 24.39 ± 4.33
Müller, 2019	[66]	Switzerland	CLBP: 141 FBSS (*n* = 44) No FBSS (*n* = 97)	CLBP: FBSS: 60.7 ± 14.2 CLBP: No FBSS: 61.3 ± 13.7	CLBP: FBSS: 21 (48%) CLBP: No FBSS: 60 (62%)	CLBP: FBSS: 29.3 ± 4.6 CLBP: No FBSS: 27.8 ± 4.4
Neblett, 2017, #	[67]	United States of America	CLBP only: 322 CLBP+: 323	CLBP only: 47.27 ± 10.56 CLBP+: 45.96 ± 11.05	CLBP only: 97 (30.1%) CLBP+: 121 (37.5%)	NR
Noord van der, 2018, #	[68]	The Netherlands	CLBP only: 19 CLBP+: 76	CLBP only: 47.58 ± 15.95 CLBP+: 45.26 ± 13.73	CLBP only: 10 (52.6%)CLBP+: 49 (64.5%)	NR
Serrano-Ibáñez, 2020, #	[69]	Spain	CLBP: 23	CLBP: 52.48 ± 10.40	CLBP: 17 (73.9%)	NR
Sharma, 2020, #	[70]	Nepal	CLBP only: 22 CLBP+: 27	CLBP only: 34.36 ± 9.88 CLBP+: 36.22 ± 13.74	CLBP only: 13 (59.1%) CLBP+: 16 (59.3%)	NR
Smart, 2012, #	[71]	Ireland and United Kingdom	CLBP only: 207 CLBP+: 134	CLBP only: 44.43 ± 14.41 CLBP+: 46.40 ± 13.07	CLBP only: 118 (57%) CLBP+: 75 (56%)	NR
Tesarz, 2015	[72]	Germany	CLBP: 149 nsCLBP-TE: (*n* = 56) nsCLBP-W-TE: (*n* = 93) Healthy controls: 31	CLBP: nsCLBP-TE: 55.8 (95% CI: 53.1; 58.6) CLBP: nsCLBP-W-TE: 58.2 (95% CI: 56.3; 60.2) Healthy controls: 60.1 (95% CI: 55.7; 64.5)	CLBP: nsCLBP-TE: 42 (75.0%) CLBP: nsCLBP-W-TE: 61 (65.6%) Healthy controls: 18 (58.1%)	CLBP: nsCLBP-TE: 29.0 (95% CI: 27.2; 30.9) CLBP: nsCLBP-W-TE: 28.2 (95% CI: 26.9; 29.5) Healthy controls: 26.8 (95% CI: 25.3; 28.2)
Tesarz, 2016	[73]	Germany	CLBP: 176 Healthy controls: 27	CLBP: 56.7 ± 10.0 Healthy controls: 57.1 ± 11.7	CLBP: 128 (72.7%) Healthy controls: 17 (63.0%)	NR

Legend: CLBP patients are categorised into CLBP only and CLBP with other pain conditions (CLBP+) when possible. When no distinction can be made between CLBP only and CLBP+ it states CLBP. 95% CI: 95% confidence interval. BMI: body mass index, CLBP: chronic low back pain, FBSS: Failed back surgery syndrome, NR: not reported, nsCLBP-TE: non-specific chronic low back pain—trauma exposure, nsCLBP-W-TE: non-specific chronic low back pain—without trauma exposure. * used the same population of patients in the studies. # Data provided by the authors.

**Table 2 jcm-10-05931-t002:** A risk of bias assessment based on QUADAS-2.

1st Author, Year	Risk of Bias					Applicability Concerns		
Patient Selection	What Index Test	Index Test	Reference Standard	Flow and Timing	Patient Selection	What Index Test	Index Test	Reference Standard
Ansuategui Echeita, 2020a [41]	☺	CSI	☹	☹	N/A	☺	CSI		☹
NOS	☹	NOS	
Ansuategui Echeita, 2020b [42]		CSI		☹	N/A		CSI	☺	☹
Aoyagi, 2019 [43]		PPT		☹	N/A	☺	PPT	☺	☹
CPM		CPM	☺
Aoyagi, 2020 [44]	☹	FM survey (WPI & SS)		☹	N/A	☺	FM survey (WPI & SS)	☺	☹
Ashina, 2018 [45]	☺	TTS	☺	☹	N/A	☹	TTS		☹
PPT	☺	PPT	
Bid, 2017 [46]		CSI	☺	☹	N/A	☺	CSI	☺	☹
Bilika, 2019 [47]	☺	CSI	☺	☹	N/A	☺	CSI	☺	☹
Chiarotto, 2018 [48]	☺	CSI	☺	☹	N/A	☺	CSI	☺	☹
Clark, 2018 [49]		CSI	☺	☹	N/A	☺	CSI	☺	☹
Clark, 2019 [50]		CSI	☺	☹	N/A	☺	CSI	☺	☹
Cuesta-Vargas, 2016 [51]	☺	CSI	☺	☹	N/A	☺	CSI	☺	☹
Defrin, 2014 [52]		QST allodynia	☺	☹	N/A	☺	QST allodynia	☹	☹
Dixon, 2016 [53]		SHS		☹	N/A	☺	SHS	☹	☹
Hubscher, 2014 [54]		thermal QST	☹	☹	N/A	☺	thermal QST		☹
Huysmans, 2018 [55]		CSI	☺	☹	N/A	☺	CSI	☺	☹
Ide, 2020 [56]	☹	CSI	☺	☹	N/A	☹	CSI	☺	☹
Knezevic, 2018 [57]		CSI	☺	☹	N/A	☺	CSI	☺	☹
Knezevic, 2020 [58]	☺	CSI	☺	☹	N/A	☺	CSI	☺	☹
Kregel, 2016 [59]		CSI	☺	☹	N/A	☺	CSI	☺	☹
Kregel, 2018 [60]		CSI	☺	☹	N/A	☺	CSI	☺	☹
PPT		PPT	☺
CPM		CPM	☺
Leemans, 2020 [61]		CSI	☺	☹	N/A	☺	CSI	☺	☹
Mayer, 2012 [25]		CSI	☺	☹	N/A	☺	CSI	☺	☹
McKernan, 2019 [62]		CSI		☹	N/A		CSI	☺	☹
MBM		MBM	
MPQ		MPQ	
Mehta, 2017 [63]		PPT		☹	N/A	☺	PPT		☹
CPM		CPM	
Mibu, 2019 [64]	☺	CSI	☺	☹	N/A	☺	CSI	☺	☹
PPT		PPT	
TS		TS	
Miki, 2020 [65]		CSI	☺	☹	N/A	☺	CSI	☺	☹
Müller, 2019 [66]		QST		☹	N/A	☺	QST		☹
Neblett, 2017 [67]	☺	CSI	☺	☹	N/A	☺	CSI	☺	☹
Noord, van der, 2018 [68]	☺	CSI	☺	☹	N/A	☺	CSI	☺	☹
Serrano-Ibáñez, 2020 [69]	☹	CSI		☹	N/A	☹	CSI	☺	☹
Sharma, 2020 [70]	☺	CSI	☺	☹	N/A	☺	CSI	☺	☹
Smart,. 2012 [71]		N/A	☹	☹	N/A		N/A	☹	☺
Tesarz, 2015 [72]		QST		☹	N/A	☺	QST		☹
Tesarz, 2016 [73]		QST		☹	N/A	☺	QST		☹

☹ High risk of bias, ☺ = Low risk of bias, 

 = Unknown, CPM: conditioned pain modulation, CSI: central sensitisation inventory, FM-survey: Fibromyalgia survey, MBM: Michigan Body Map—revised version, MPQ: McGill Pain Questionnaire—short form-revised, N/A: not applicable. NOS: waddle non-organic signs, PPT: pressure pain threshold, QST: quantitative sensory testing, SHS: Sensory Hypersensitivity Scale, SS: symptom severity, TS: temporal summation, TTS: total tenderness score, WPI: widespread pain index.

**Table 3 jcm-10-05931-t003:** Definition used to describe human assumed central sensitisation and the reported prevalence of human assumed central sensitisation.

1st Author, Year	Definition of HACS or HACS Similar Definition	Reference Definition HACS (1st Author, Year)	Prevalence HACS in Patients with CLBP Stated in the Article	CSI	
				Mean	Prevalence (Cut-Off CSI 40)
Ansuategui Echeita, 2020a [41] #	“Central Sensitisation was introduced as a possible pathophysiological mechanism in several chronic pain conditions, including a subgroup of patients with CBP.”	Woolf, 1983 [16] Roussel, 2013 [13]	NR	34.7 ± 13.1	22 out of 56 (39.3%)
Ansuategui Echeita, 2020b [42] #	“In a subgroup of patients with chronic pain, pain might not be direct reflection of the presence of a noxious peripheral stimulus (nociceptive pain) nor the nervous system (neuropathic pain), but could be the result of a condition in which the CNS is in a hypersensitive state; central sensitisation.”	Woolf, 2011 [18]			
Aoyagi, 2019 [43]	“Defined as augmented central pain processing.”	Woolf, 2007 [77] Latremoliere, 2009 [78] Woolf, 2011 [18] Clauw, 2015 [79] Nijs, 2014 [80] Roussel, 2013 [13]	NR	NA	NA
Aoyagi, 2020 [44]	“Defined as amplified pain processing in the central nervous system.”	Clauw, 2015 [79] Nijs, 2015 [81] Roussel, 2013 [13]	NR	NA	NA
Ashina, 2018 [45] #	“Both back pain and primary headache disorders may play a role in the sensitisation of partially overlapping central nociceptive pathways.”	Yoon, 2013 [82]	NR	NA	NA
Bid, 2017 [46]	“CS is described by the International Association for the Study of Pain (IASP) as: "Increased responsiveness of nociceptive neurons in the central nervous system to their normal or subthreshold afferent input". CS is also defined as "an augmentation of responsiveness of central neurons to input from unimodal and polymodal receptors".”	Loeser, 2008 [7] Meyer, 1995 [83]	Experimental (*n* = 64): 78.1% Control (*n* = 64): 64,1% Based on the CSI	Baseline Experimental: 45.68 Control: 37.34 Week 4 Experimental: 23.42 Control: 28.21 Week 8 Experimental: 11.17 Control: 21.17	91 out of 128 (71.1%)
Bilika, 2020 [47] #	“A phenomenon of hypersensitivity of the central nervous system in patients with chronic pain.”	Roussel, 2013 [13] Woolf, 2011 [18]	NR	31.79 ± 12.19	CLBP only: 9 out of 28 (32.14%) CLBP+: 1 out of 23 (4.17%)
Chiarotto 2018 [48] #	“an amplification of neural signalling within the central nervous system that elicits pain hypersensitivity”	Woolf, 2011 [18]	NR	33.93 ± 11.88	NR
Clark, 2018 [49]	“Central sensitisation involves facilitation of peripheral stimulus processing and alterations in descending inhibitory control of nociceptive input to the brain.”	Woolf, 2011 [18] Nijs, 2010 [76] Mayer, 2012 [25]	NR	46.14 ± 19.39	16 out of 21 (76.2%)
Clark, 2019 [50] #	“A dysregulation of the central nervous system causing neuronal hyperexcitability, characterized by generalized hypersensitivity of the somatosensory system to both noxious and non-noxious stimuli.”	Nijs, 2010 [76] Mayer, 2012 [25] Neblett, 2013 [84]	NR	50.10 ± 13.86	125 out of 165 (75.8%)
Cuesta-Vargas, 2016 [51] #	“CS involves an abnormal increase of pain caused by neuronal hyperexcitability and dysfunction in descending and ascending pathways in the central nervous system.”	Kindler, 2011 [85] Heinricher, 2009 [86]	NR	CLBP only: 22.57 ± 11.37 CLBP+: 25.62 ± 12.22	CLBP only: 7 out of 107 (6.5%) CLBP+: 7 out of 73 (9.6%)
Defrin, 2014 [52]	“Current pain theory holds that sustained peripheral noxious input, whether due to sensitized sensory endings or ectopic pacemaker activity, may secondarily initiate a state of spinal central sensitisation. In this state, afferent input is amplified and activity in low threshold Ab mechanosensitive afferents is rendered painful (Ab pain). A well-known example is secondary hyperalgesia, a region of hypersensibility to light touch (tactile allodynia) on the skin that surrounds the location of a primary noxious input.”	Raja, 1984 [87] Torebjörk, 1992 [88] Woolf, 2011 [18]	CLBP+: 60.8%, based on the presence of tactile allodynia CLBP only: 13.3%	NA	NA
Dixon, 2016 [53]	“Central sensitisation is an amplified state of neural signalling in the central nervous system (CNS) that is implicated in the pathogenesis of several chronic conditions that primarily involve pain and complex, multisymptom illnesses. When in the sensitized state, the CNS amplifies the sensory processing of the peripheral inputs so that the experience of the individual no longer accurately reflects the information provided by peripheral inputs. This state has been described as an increase in signal gain in which low-level sensory inputs are amplified into stronger signals, or as a decrease in signal inhibition processes, or both.”	Kaya, 2013 [89] Lluch, 2014 [90] Wang, 2014 [91] Batheja, 2013 [92] Woolf, 2011 [18]	NR	NA	NA
Hubscher, 2014 [54]	“Parallel to this peripheral phenomenon, intense ongoing peripheral nociceptive input can lead to altered central mechanisms, such as, an immediate-onset and lasting increase in the excitability of dorsal horn pain transmission neurons, referred to as central sensitisation. Central sensitisation may manifest as pain hypersensitivity (eg, allodynia, hyperalgesia, temporal summation [TS]) that can spread to non-injured areas.”	Ji, 2003 [93] Salter, 2004 [94] Woolf, 2011 [18]	NR	NA	NA
Huysmans, 2018 [55]	“Central sensitisation can be defined as a process of abnormal and intense enhancement of pain caused by increased neuronal responses to stimuli in the central nervous system. This central hyperexcitability is associated with altered sensory processing in the brain, malfunctioning of endogenous pain inhibitory systems, increased activity of pain facilitatory pathways, and temporal summation of second pain and/or wind-up, which leads to dysfunctional endogenous analgesic control.”	Nijs, 2015 [81] Mayer, 2012 [25] Yunus, 2007 [95] Nijs, 2010 [76] Nijs, 2011 [96] Woolf, 2011 [18] Staud, 2007 [97] Meeus, 2008 [98] Meeus, 2007 [99]	NR	32.92 ± 12.76 (range: 16–66)	12 out of 38 (31.6%)
Ide, 2020 [56] #	“The International Association for the Study of Pain defines central sensitisation (CS) as “increased responsiveness of nociceptive neurons in the central nervous system to their normal or subthreshold afferent input”.”	Loeser, 2008 [7]	NR	CLBP only: 7.76 ± 6.43 CLBP+: 17.77 ± 9.93	CLBP only: 0 out of 46 (0%) CLBP+: 4 out of 206 (1.94%)
Knezevic, 2018 [57] #	“Central sensitisation (CS) represents “increased responsiveness of nociceptive neurons in the central nervous system to their normal or subthreshold afferent input.” Peripheral stimuli that are otherwise innocuous can produce augmented, prolonged, and widely spread pain.”	International Association for the Study of Pain, 2012 [100] Woolf, 2011 [18]	NR	CLBP only: 36.94 ± 16.15 CLBP+: 44.66 ± 14.98	CLBP only: 68 out of 157 (43.3%)
Knezevic, 2020 [58] #	“Central sensitisation refers to hypersensitivity of the central nervous system, resulting in enhancement of pain sensations.”	Woolf, 2011 [18] Mayer, 2012 [25] Neblett, 2017 [67]	NR	CLBP only: 36.42 ± 15.51 CLBP+: 44.64 ± 13.94	CLBP only: 65 out of 155 (41.9%) CLBP+: 51 out of 88 (58.0%)
Kregel, 2016 [59] #	“Central sensitisation (CS) is a neurophysiological state resulting in hyperexcitability in the central nervous system. According to Woolf, CS is “operationally defined as an amplification of neural signalling within the central nervous system that elicits pain hypersensitivity.” In clinical practice, CS manifests as pain hypersensitivity, particularly dynamic tactile allodynia, secondary punctate or pressure hyperalgesia, longer aftersensations, and enhanced temporal summation.”	Woolf, 2011 [18] Nijs, 2010 [76]	NR	CLBP only: 23.67 ± 10.50 CLBP+: 38.90 ± 14.77	CLBP only: 1 out of 4 (25.0%) CLBP+: 7 out of 11 (63.6%)
Kregel, 2018 [60]	“Dysregulations of ascending and descending pathways have been observed in chronic pain patients, resulting in clinical signs such as allodynia, hyperalgesia, hypersensitivity, increased or prolonged aftersensations, and temporal summation to noxious and non-noxious stimuli. Extended high-frequency stimulation of neurons has been found to cause long-lasting cellular changes because of elevated cell responsiveness, a diminished working of the inhibitory cells and network sprouting. This increase in excitability and synaptic working in the central nociceptive pathways is called central sensitisation.”	Woolf, 2011 [18] Schliessbach, 2013 [101] Baranauskas, 1998 [102] Nijs, 2015 [81] Lluch, 2014 [90] Maixner, 1998 [103] Wilgen, van, 2013 [104]	NR	CLBP: 39.06 ± 11.61	NR
Leemans, 2020 [61]	NR	NA	NR	CLBP: Experimental group: 35.9 ± 10.5 CLBP: Control group: 31 ± 10.8	NR
Mayer, 2012 [25]	*In the abstract:* “Central sensitisation has been proposed as a common pathophysiological mechanism to explain related syndromes for which no specific organic cause can be found.” *In the introduction:* “Central sensitisation, which involves an abnormal and intense enhancement of pain by mechanisms in the central nervous system, maybe the common link between these disorders.”	Yunus, 2007 [95]	NR	CLBP only: 41.6 ± 14.8	NR
McKernan, 2019 [62] #	“Central sensitisation—the amplification of neural signalling in the central nervous system contributing to hyperalgesia.”	Woolf, 2011 [18]	NR	CLBP only: 50.83 ± 16.67	NR
Mehta, 2017 [63]	“Central sensitisation; this may manifest as pain hypersensitivity, in particular dynamic tactile allodynia, secondary punctate or pressure hyperalgesia, and enhanced temporal summation. Central sensitisation is a hyperexcitability state in nociceptive pathways and has been suggested to be the main cause of chronic pain conditions.”	NR	NR	NA	NA
Mibu, 2019 [64]	“The International Association for the Study of Pain defines central sensitisation as an increased responsiveness of nociceptive neurons in the central nervous system to normal or subthreshold afferent input.”	Loeser, 2008 [7]	*n* = 104: 19 (18.3%) Based on PPT and TS	CLBP: 25.5 ± 12.2	NR
Miki, 2020 [65]	“Central sensitisation is defined by the International Academy of Pain as a functional dysregulation of the central nervous system to normal or subthreshold afferent input. The nociceptive hyperexcitability and perception threshold of sensory information are reduced, and pain and other clinical symptoms are amplified.”	Loeser, 2008 [7] Woolf, 1983 [16]	NR	CLBP: 24.44 ± 12.78	31 out of 238 (13.0%)
Müller, 2019 [66]	“Central hypersensitivity: Prolonged or intense nociceptive input induces neuroplastic changes that lead to central nervous system hypersensitivity.“	Woolf, 2011 [18]	NR	NA	NA
Neblett, 2017 [67] #	“Central sensitisation is a relatively new concept, which is gaining wide acceptance as a functional dysregulation in the central nervous system, resulting in nociceptive hyperexcitability and a lowered threshold for perception of sensory information, which amplifies pain and other clinical symptoms.”	Adams, 2015 [105]	NR	CLBP only: 44.21 ± 15.24 CLBP+: 49.24 ± 15.01	CLBP only: 200 out of 322 (62.1%) CLBP+: 237 out of 323 (73.4%)
Noord van der, 2018 [68] #	“Central sensitisation is a common neurophysiological phenomenon in patients with chronic pain. Central sensitisation involves a hyperexcitability to a stimulus, resulting in an abnormal response to both noxious and non-noxious stimuli.”	Schliessbach, 2013 [101] Woolf, 2011 [18]	NR	CLBP only: 29.41 ± 14.03 CLBP+: 40.55 ± 14.28	CLBP only: 4 out of 17 (23.5%) CLBP+: 32 out of 67 (47.8%)
Serrano-Ibáñez, 2020 [69] #	“The International Association of the Study of Pain has defined central sensitisation as the increased responsiveness of nociceptive neurons in the central nervous system to normal or subthreshold afferent input.”	Loeser, 2008 [7]	NR	CLBP: 63.68 ± 13.57	CLBP: 16 out of 24 (66.7%)
Sharma, 2020 [70] #	“Central sensitisation involves the amplification of pain, and hypersensitivity to other environmental stimuli, within the central nervous system.”	Woolf, 2011 [18]	NR	CLBP only: 24.27 ± 13.12 CLBP+: 24.00 ± 12.53	CLBP only: 3 out of 22 (14.8%) CLBP+: 4 out of 27 (13.6%)
Smart, 2012 [71] #	“Central sensitisation pain (CSP) refers to pain that arises or persists as a result of aberrant processing and/or hypersensitivity within the diffuse neural networks of the central nervous system (CNS) engaged in nociception, in the absence of or disproportionate to somatic tissue or peripheral nerve pathology.”	Costigan, 2009 [106]	NR	NR	NR
Tesarz, 2015 [72]	NR	NA	NR	NA	NA
Tesarz, 2016 [73]	NR	NA	NR	NA	NA
		Total	50.65%	All	1013 out of 2347 (43.2%)
				CLBP only	289 out of 701 (41.2%)
				CLBP+	343 out of 819 (41.9%)

Legend: CBP: chronic back pain, CLBP: Chronic low back pain, CLBP+: patients with chronic low back pain in combination with other pain condition(s), CNS: central nervous system, CS: central sensitisation, CSI: central sensitisation inventory, CSP: central sensitisation pain. HACS: human assumed central sensitisation, IASP: International Association for the Study of Pain, NA: not applicable, NR: not reported, PPT: pressure pain threshold, TS: temporal summation, # Data provided by the authors.

**Table 4 jcm-10-05931-t004:** Reported assessment of Human Assumed Central Sensitisation in patients with chronic low back pain.

Questionnaires				
CSI (*n* = 23)				
Study (1st author, year)	The goal of the test	Clinimetrics	Comparison between assessment methods	
Ansuategui Echeita, 2020a [41]	Quantify the severity of symptoms CS	Not reported	CSI with Waddle Non-organic Signs.	
Ansuategui Echeita, 2020b [42]	Quantify the severity of symptoms CS	Not reported	CSI with Lifting capacity	
Bid, 2017 [46]	A score above 40 indicates the presence of CS	Not reported	Comparing CSI (CS group/NoCS group) with PPT scores, numeric pain rating scale, Roland Morris Disability Questionnaire, Fear-Avoidance Beliefs Questionnaire, Trunk Flexors Endurance, and Trunk Extensor Endurance	
Bilika, 2020 [47]	Identify symptoms associated with CS	Internal consistency: Cronbach’s α = 0.994 Test-retest: ICC = 0.993	CSI with pain catastrophizing scale.	
Chiarotto, 2018 [48]	Identify patient’s symptoms related to CS	Internal consistency: Cronbach’s α = 0.87	No comparison	
Clark, 2018 [49]	Person’s symptoms likely to be attributable to CS	Not reported	CSI (CSI High group/CSI Low group) with Sensory Seeking, Sensory Sensitive, trait anxiety, Low Registration, and Sensation Avoidance.	
Clark, 2019 [50]	Individual’s symptoms likely to be attributable to CS	Not reported	CSI with sensory profiles, Sensory Sensitivity, sensation avoiding, low registration, sensation seeking, and trait anxiety.	
Huysmans, 2018 [55]	The degree of symptoms of CS	Not reported	CSI and 1-minute stair-climbing test, Pain catastrophizing scale, visual analogue scale at this moment, Brief Illness Perception Questionnaire, Quebec Back Pain Disability Scale, and Tampa Scale for Kinesiophobia.	
Ide, 2020 [56]	Assessing CS syndrome (CSS)	Not reported	CSI and EuroQOL 5-dimension, Neck Disability Index, and Oswestry Disability Index.	
Knezevic, 2018 [57]	Assesses 25 symptom dimensions associated with CS and CSS.	Internal consistency: Cronbach α = 0.909 Test-retest: ICC = 0.947	No comparison	
Knezevic, 2020 [58]	A measure of symptoms related to CS and CSS	Not reported	CSI with Medical Outcomes Study, Fear-Avoidance Components Scale, Oswestry Disability Index, Short Form-36, Pain Catastrophizing Scale, pain intensity, and Multidimensional Scale of Perceived Social Support.	
Kregel, 2016 [59]	Measure the overlapping symptom dimensions present in CS.	Internal consistency: Cronbach α= 0.91 Test-retest: ICC = 0.88	No comparison	
Kregel, 2018 [60]	An indirect tool for CS symptomatology evaluation	Not reported	CSI with PPT, CPM, current pain intensity, quality of life, pain disability, and pain catastrophizing score	
Leemans, 2020 [61]	Identify key symptoms associated with CS	Not reported	No comparison	
Mayer, 2012 [25]	Assess symptoms associated with CS	Internal consistency: Cronbach α = 0.879. Test-retest: ICC = 0.817	No comparison	
McKernan, 2019 [62]	Assess key polysomatic symptoms associated with a CS disorder	Not reported	CSI with Trauma History Questionnaire, PTSD, Michigan Body Map, McGill Pain Questionnaire, Multidimensional Experiential Avoidance Questionnaire.	
Mibu, 2019 [64]	Assess health-related symptoms in CSS	Sensitivity: CS+ ^1^ or CS− ^1^: 73.7% (cut-off: 20) CSS+ ^2^ or CSS− ^2^: 69.2% (cut-off: 28) Specificity: CS+ ^1^ or CS− ^1^: 37.7% CSS+ ^2^ or CSS− ^2^: 69.2%	CSI and duration of symptoms, EQ-5D, pain intensity, pain interference, Widespread Pain Index score, PPT, and temporal summation.	
Miki, 2020 [65]	Significant deficits in CS	Not reported	CSI (low CSI group/high CSI group) with pain catastrophizing scale, Tampa Scale for Kinesiophobia, Hospital Anxiety and Depression Scale, pain intensity for LBP, pain intensity for leg pain, Roland Morris Disability Questionnaire, and EuroQoL 5 dimensions.	
Neblett, 2017 [67]	Screener for high risk of having CSS	Not reported	Explored the five CSI severity levels with patient-reported outcomes: for pain intensity, perceived disability, depressive symptoms, sleep disturbance, pain-reported outcomes; pain intensity, perceived disability, depressive symptoms, sleep disturbance, pain-related anxiety, and somatization-related symptoms.	
Noord, van der, 2018 [68]	Identifying symptoms of CS in patients with chronic pain disorders	Not reported	CSI part A with CSI part B, depression, anxiety, WPI, pain intensity, and pain catastrophizing scale.	
Serrano-Ibáñez, 2020 [69]	Severity of CS	Not reported	CSI with daily routines, decreased physical activity, diminished social support, emotional distress, and pain intensity.	
Sharma, 2020 [70]	Assess somatic and emotional health-related symptoms associated with CS	Internal consistency: Cronbach’s α = 0.87 ICC = 0.98 (95% CI: 0.97, 0.99)	CSI with the pain catastrophizing scale (strong correlation), number of pain descriptors(McGill Pain Questionnaire) (moderate correlation), and pain intensity (weak correlation)	
**MBM (*n* = 1)**				
**Study (1st author, year)**	**The goal of the test**	**Clinimetrics**	**Version**	**Comparison**
McKernan, 2019 [62]	Indicate widespread pain related to CS	Not reported	Revised version [107]	Exposure to trauma and PTSD increases CS. Findings need to be objectified with laboratory markers of CS.
**MPQ (*n* = 1)**				
**Study (1st author, year)**	**The goal of the test**	**Clinimetrics**	**Version**	**Comparison**
McKernan, 2019 [62]	Assessing various dimensions of pain (Indicator for CS)	Not reported	SF-MPQ-2 [108,109]	Exposure to trauma and PTSD increases CS. Findings need to be objectified with laboratory markers of CS.
**WPI (*n* = 2)**				
**Study (1st author, year)**	**The goal of the test**	**Clinimetrics**	**Version**	**Comparison**
Aoyagi, 2019 [43]	Assesses experience pain or tenderness in 19 specific body areas. As a continuous variable to measure CS severity	Not reported	as part of the 2011 FM survey [110,111]	FM positive when WPI ≥ 7 and ≥ 5 or WPI 3–6 and SS ≥ 9. Conclusion article: FM positive = CS
Aoyagi, 2020 [44]	Scores from the WPI and SS are combined to determine the presence and severity of CS.	Not reported	As part of the 2011 FM survey [110,111]	Cutoff scores of ≥ 12 with a combination of either WPI score ≥ seven and SS score ≥ five or WPI score 3 to 6 and SS score ≥ 9 distinguish those with CS as FM positive. Higher total scores indicate a greater degree of CS.
**QST measurements**				
**PPT (*n* = 7)**				
**Study (1st author, year)**	**The goal of the test**	**Clinimetrics**	**Method /location(s)**	**Comparison**
Aoyagi, 2019 [43]	Identifying individuals with CS	Not reported	Handheld algometer *Thumbnail* *Lower back*	PPT values were compared between the FM-negative and FM-positive group. FM scores were used as a dichotomous variable to identify the presence of CS and as a continuous variable to examine associations between CS, QST and other self-reported measures.
Aoyagi, 2020 [44]	Identifying individuals with CS	Not reported	Handheld algometer *Thumbnail* *Lower leg*	PPT values were compared between the FM-negative and FM-positive group.
Kregel, 2018 [60]	To objectify CS symptomatology/evaluation of CS symptoms	Not reported	Handheld algometer *Lower back* *Hand* *Upper leg*	The CSI compared with measures of pain intensity, quality of life, pain disability, pain catastrophizing, PPT, and CPM
Leemans, 2020 [61]	Altered sensory processing, including signs of CS	Not reported	Handheld algometer *Three spots in the lower back* *2nd Toe*	No conclusions about CS
Mibu, 2019 [64]	The lowest tertile PPT, in combination with a positive TS, are patients with CS.	Not reported	Handheld algometer *Lower arm*	No comparison
Tesarz, 2015 [72]	It covers all relevant aspects of the somatosenosory system, including large and small fibre functions and signs of central sensitisation.	Not reported	Handheld algometer *Low back* *Dorsum hand*	No comparison
Tesarz, 2016 [73]	It covers all relevant aspects of the somatosenosory system, including large and small fibre functions, and signs of central sensitisation	Not reported	Handheld algometer *Low back* *Dorsum hand*	No comparison
**CPM (*n* = 3)**				
**Study (1st author, year)**	**The goal of the test**	**Clinimetrics**	**Method/location(s)**	**Comparison**
Aoyagi, 2019 [43]	Discriminate individuals with CS	Not reported	PPT before and after. Conditioning painful stimulus cuff to ischemic pain. *Thumbnail* *Lower back*	PPT values were compared between the FM-negative and FM-positive group. FM scores were used as a dichotomous variable to identify the presence of CS and as a continuous variable to examine associations between CS, QST and other self-reported measures.
Kregel, 2018 [60]	To objectify CS symptomatology/evaluation of CS symptoms	Not reported	Cold Pressor Test. 1 min. 22 °C, 2 min. 12 °C, 30 s. wait, PPT measurements *Upper leg*	The CSI compared with measures of pain intensity, quality of life, pain disability, pain catastrophizing, PPT, and CPM
Leemans, 2020 [61]	Altered sensory processing, including signs of CS, to evaluate the efficacy of the descending inhibitory modulation of pain	Not reported	Cold pressor test. 0.7 °C until intolerable or 2 min. PPT before and after *2nd Toe*	No comparison
**TS (*n* = 4)**				
**Study (1st author, year)**	**The goal of the test**	**Clinimetrics**	**Method/ location(s)**	**Comparison**
Hubscher, 2014 [54]	Thermal pain thresholds and tolerance (heat/cold) and TS of heat pain. The distal site as a marker of possible CS.	Not reported	One sequence of 10 consecutive heat pulses of <1-s duration at an interstimulus interval of 0.33 Hz was delivered. The temperature increased from 41 °C to a maximum of 47 °C at a rate of 10 °C/3. The pain intensity of each heat pulses was assessed. *Location:* 2 sites: on the surface of the low back and a distal site, the volar surface of the forearm	No comparison
Mibu, 2019 [64]	The lowest tertile PPT, in combination with a positive TS, are patients with CS.	Not reported	Previous determined PPT was applied ten times *Lower arm*	No comparison
Tesarz, 2015 [72]	It covers all relevant aspects of the somatosenosory system, including large and small fibre functions, and signs of central sensitisation	Not reported	Pinprick 256N *Low back* *Dorsum hand*	No comparison
Tesarz, 2016 [73]	It covers all relevant aspects of the somatosenosory system, including large and small fibre functions and signs of central sensitisation.	Not reported	Pinprick 256N *Low back* *Dorsum hand*	No comparison
**Thermal QST (*n* = 3)**				
**Study (1st author, year)**	**The goal of the test**	**Clinimetrics**	**Method/Location(s)**	**Comparison**
Hubscher, 2014 [54]	Thermal pain thresholds and tolerance (heat/cold) and TS of heat pain. The distal site as a marker of possible CS.	Not reported	CPT, CPTol, HPT, HPTol. *2 sites: on the surface of the low back and a distal site, the volar surface of the forearm*	No comparison
Tesarz, 2015 [72]	All relevant aspects of the somatosenosory system, including large and small fibre functions and signs of central sensitisation.	Not reported	TSA 2001-II CDT, WDT, TSL, CPT, HPT. *Low back* *Dorsum hand*	No comparison
Tesarz, 2016 [73]	All relevant aspects of the somatosenosory system, including large and small fibre functions and signs of central sensitisation.	Not reported	TSA 2001-II CDT, WDT, TSL, CPT, PHS, HPT. *Low back* *Dorsum hand*	No comparison
**Other QST measures (*n* = 2)**				
**Study (1st author, year)**	**The goal of the test**	**Clinimetrics**	**Method/Location(s)**	**Comparison**
Tesarz, 2015 [72]	All relevant aspects of the somatosenosory system, including large and small fibre functions and signs of central sensitisation	Not reported	MPT (Pinprick stimulators), MPS (Pinprick stimulators), DMA (brush, cotton wool and Q-tip), MDT (von Frey filaments), VDT (tuning fork 64 Hz) *Low back* *Dorsum hand*	No comparison
Tesarz, 2016 [73]	It covers all relevant aspects of the somatosenosory system, including large and small fibre functions and signs of central sensitisation.	Not reported	MPT (Pinprick stimulators), MPS (Pinprick stimulators), DMA (brush, cotton wool and Q-tip), MDT (von Frey filaments), VDT (tuning fork 64 Hz) *Low back* *Dorsum hand*	No comparison
**No measurements (*n* = 6)**				
**Study (1st author, year)**	**The goal of the test**	**Clinimetrics**	**Explanation of possible HACS measures**	
Ashina, 2018 [45]	Not reported	Not reported	Discussion section: lower cephalic and extra-cephalic PPT and higher TTS in the chronic headache group than episodic headache and control groups suggest that comorbidity of back pain and frequent headaches is associated with signs of CS. TTS is increased, suggesting that low back pain can induce CS.	
Defrin, 2014 [52]	Not reported	Not reported	Results section: the development of tactile allodynia and inference of CS has more to do with individual predisposition than with the intensity of the precipitating noxious input. Discussion section: Neural mechanism: CS and ectopic hyperexcitability. The presence of tactile allodynia strongly implies the presence of CS. The observed ~60% incidence of leg allodynia in radicular patients suggests that peripheral nervous system generators of leg pain often induce CS. The 40% who did not (yet) develop CS despite comparable leg pain were presumably less prone to doing so.	
Dixon, 2016 [53]	Not reported	Not reported	CS is used as an explanatory model of the results	
Mehta, 2017 [63]	Not reported	Not reported	Changes in PPT and CPM are consistent with normalization of peripheral and CS	
Müller, 2019 [66]	Not reported	Not reported	Negative findings for QST as a predictor for FBSS. They conclude that the negative findings do not necessarily mean that central hypersensitivity is not involved in FBSS.	
Smart, 2012 [71]	Not reported	Not reported	Based on clinical examination, patients were, i.e., CS	

Abbreviations: CDT, cold detection threshold, CI: confidence interval, CPM: conditioned pain modulation, CPT, cold pain threshold, CPTol: cold pain tolerance, CS: central sensitisation, CSI: central sensitisation inventory, CSS: central sensitisation syndromes, DMA, dynamic mechanical allodynia, FBSS: Failed back surgery syndrome, FM: fibromyalgia, HPT, heat pain threshold, HPTol: heat pain tolerance, ICC: intraclass correlation, LBP: low back pain, MBM: Michigan Body Map, MDT, mechanical detection threshold, MPQ: McGill Pain Questionnaire, MPS, mechanical pain sensitivity, MPT, mechanical pain threshold, PHS: paradoxical heat sensations, PPT, pressure pain threshold, PPT: pressure pain threshold, PTSD: post-traumatic stress disorder, QST: quantitative sensory testing, SS: symptom severity, TS: Temporal summation, TSA: Advanced thermosensory stimulator, TSL, thermal sensory limen, TTS: total tenderness score, VDT, vibration detection threshold, WDT, warm detection threshold, WPI: Widespread Pain Index, WUR, wind-up ratio,. ^1^: CS was determent based on the lowest tertile of the PPT data and positive values of TS. ^2^: CSS was determent based on the number of CSS in the CSI part B.

**Table 5 jcm-10-05931-t005:** Substantial correlations (correlation > 0.5) between the measures used to assess HACS and other assessments.

			CLBP Only	CLBP+
Author, Year	Assessment	Type	CSI Part A	
Disability				
Ansuategui Echeita, 2020b [42]	Lifting capacity	PA	−0.53	
Ide, 2020 [56] #	Neck Disability Index	Q	0.58	0.60
Ide, 2020 [56] #	Oswestry Disability Index ᵕ	Q		0.50
Kregel, 2018 [60]	Physical components (Short Form-36) ᵕ	Q	−0.62	
**Pain**				
Huysmans, 2018 [55]	Pain Score VAS: 7 days	Q	0.51	
Huysmans, 2018 [55]	Pain Score VAS: now	Q	0.51	
McKernan, 2019 [62]	McGill Pain Questionnaire ᵕ	Q	0.62	
McKernan, 2019 [62]	Michigan Body Map	Q	0.55	
Serrano-Ibáñez, 2020 [69] #	NRS pain intensity ᵕ	Q	0.60	
**Co-morbidities**				
Van der Noord, 2018 [68]	Central sensitivity syndrome	Q	0.51	
**Psychological elements**				
Bilika, 2020 [47] #	Pain Catastrophizing Scale (total score) ᵔ ᵕ	Q	0.74	0.56
Clark, 2018 [49]	Sensory profile: Sensory seeking ᵕ	Q	−0.53	
Clark, 2018 [49]	Sensory profile: Sensory Sensitive ᵔ	Q	0.57	
Clark, 2018 [49]	State-Trait Anxiety Inventory ᵕ	Q	0.63	
Clark, 2019 [50]	Sensory profile: Low registration ᵕ	Q	0.54	
Clark, 2019 [50]	Sensory profile: Sensory Sensitive ᵔ	Q	0.63	
Huysmans, 2018 [55]	Pain Catastrophizing Scale (total score) ᵔ ᵕ	Q	0.52	
Kregel, 2018 [60]	Mental components (Short Form-36) ᵕ	Q	−0.64	
McKernan, 2019 [62]	PTSD (PCL)	Q	0.65	
Miki, 2020 [65]	Anxiety (Hospital Anxiety and Depression Scale)	Q	0.50	
Miki, 2020 [65]	Pain Catastrophizing Scale (total score) ᵔ ᵕ	Q	0.54	
Serrano-Ibáñez, 2020 [69] #	Emotional distress	Q	0.56	
Sharma, 2020 [70]	Pain Catastrophizing Scale (total score) ᵔ ᵕ	Q	0.50	
Van der Noord, 2018 [68]	Anxiety (SCL-90)	Q	0.65	
Van der Noord, 2018 [68]	Depression (SCL-90)	Q	0.67	
**Sleep**				
Knezevic, 2020 [58] #	Sleep problem Index II (MOS sleep scale)	Q	−0.52	

Note: McGill Pain Questionnaire and PTSD (PCL) were correlated 0.51 [62] For the full table see Appendix A. Legend: PA: physical assessment, Q: questionnaire, VAS: visual analogue scale, MOS: medical outcomes study, SCL-90: Symptom checklist. # Data provided by the authors. ᵔ Assessment multiple times in this table. ᵕ Assessment has also correlations below 0.5, see Appendix A.

**Table 6 jcm-10-05931-t006:** Indicator tests for the grading system for human assumed central sensitisation in patients with chronic low back pain.

Indicator Tests		
Hypersensitivity	Pressure pain threshold (PPT)	
Temporal summation	Wind-up ratio (WUR)	Positive WUR
Reduced pain inhibition	Conditioned pain modulation (CPM)	Negative CPM
Questionnaire	Central Sensitisation Inventory (CSI)	Score ≥ 40

## Data Availability

The authors of the original papers can request all data and related metadata underlying this review’s findings. Data can also be requested from the first author of this systematic review, but data will only be provided with the original data authors consent.

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
