# Peer review of "The Definition, Assessment, and Prevalence of (Human Assumed) Central Sensitisation in Patients with Chronic Low Back Pain: A Systematic Review"

_jcm, 2021, doi:10.3390/jcm10245931_

Round 1
Reviewer 1 Report
See the attached document.

Author Response
See the attached document
The definition, assessment, and prevalence of (Human Assumed) Central Sensitisation in patients with chronic low back pain. A systematic review
Manuscript ID jcm-1454927
Response to Reviewer 1 Comments
Thank you for your helpful feedback and insights. We elaborated your suggestions and comments by discussion within the group. According to this we made the following changes to the manuscript.
The authors have conducted an systematic review about three aspects of Central sensitization mechanism in the chronic low back pain population: definition, assessment and prevalence. Furthermore, they propose a new term for central sensitization when used with human population, attempting to fix a taxonomy problem: the use in humans of a term of a mechanism that cannot be directly measured in humans is problematic. Therefore, they proposed “Human assumed central sensitization (HACS)”. The work done is very valuable and necessary, but in my view, the paper rises important and interesting epistemological and taxonomical challenges. Therefore, please, take this review as an acknowledgement to your work.
Introduction
- P2, Line 64 “There is an apparent conceptual overlap between CS and nociplastic pain, however the proposed underlying (patho)physiological mechanisms may differ”. Different pathophysiological mechanisms can´t be the reason explaining the differences between CS and nociplastic pain. It should not be assumed that they belong to the same category, they are just different entities: central sensitization is a change in the physiology of central sensory neurons, and despite it is a mechanistic descriptor, (nociplastic)PAIN, since it is pain, is a subjective experience (not a thing). Pears can´t be compared with apples. Please, clarify this in the text.
Response 1: Thank you for your comment. The distinction should be made more clearly and we agree your comment. Therefore we revised this section into “It has been suggested that there is an apparent conceptual overlap between CS and nociplastic pain, yet the terms stand for different entities. [19]. CS refers to a neural mechanism and nociplastic pain refers to a pain mechanism. However both CS and nociplastic pain supposedly have altered nociception, which could originate either peripherally or centrally.” P2, Line 64-69.
- P2, Line 75: Considering that taxonomy is key in this review, and that the authors have done an effort trying to improve the use of the CS as a term, they should be careful with terminology and its consistency. Is it correct to say, “CS syndromes”? Even if we could directly assess CS in humans, CS wouldn´t be a diagnosis. They authors should be consistent regarding the proper terms addressing chronic pain populations without objective tissue damage: chronic pain patients, patients with nociplastic pain, central sensitization syndromes (better not to use this one), complex pain, centralised pain…etc. Please, decide and be consistent throughout the text.
Response 2: Thanks for bringing this up. We followed your suggestion and changed ‘CS syndromes’ and ‘CSS’ into ‘syndrome/disorder/disease related to HACS’ throughout the manuscript but not in the tables, because these where direct citations from the included papers. We changed the sentence about CS syndromes (P3, Line 74-75) into ”… most humans with syndrome/disorder/disease related to assumed CS” because HACS has not been introduced yet.
- The HACS term proposal: I see more limitations in this new term than those proposed by the authors. Despite I agree with the authors on the need to improve the concept of CS in humans, I am still sceptical regarding the use of HACS:
a. Central sensitization is relatively simple: Clifford Woolf generated first formal evidence of CS in 1983 by injuring rats and observing how the response properties of dorsal horn neurons changed due to intense/constant nociceptive stimulation. Now we know that these properties can change also due to altered descending nociception controls. Anyways, central sensitization refers to increased response of central nervous system neurons when afferents from the periphery activate. This might cause lots of symptoms and have multiple manifestations: as many as existing assessment methods. Therefore, it is very difficult for the concept of HACS to be clearly defined or to agree on a unified definition, as it may involve many different manifestations: altered CPM, decreased pain thresholds to different stimuli, increased temporal summation, widespread pain, increased nociceptive reflex (withdrawal reflex)…In summary, CS in one ““thing””, but HACS can be many different things. Why then not including in the acronym a word referring to symptoms and manifestations, that can be assumed to be derived from CS? E.g., CSM - Central sensitization manifestations, CSS – Central sensitization symptoms. This new proposed taxonomy would better reflect reality.
Response 3a: Thank you for your insights. You suggest HACS can present itself in many ways while CS is only ‘one thing’, a neural mechanism. We agree on that and we have added this to the discussion section as being a very valuable point.
The idea behind the term HACS is that although CS is a mechanism that still cannot be demonstrated in humans, the manifestations in humans may refer to the possible presence of CS. Therefore in humans it can be assumed that CS as a neural mechanism may be present. This is why we suggest to name HACS. We think that HACS is more like CS while terms like CSM and CSS as suggested are more descriptive in nature which can be seen as expressions or manifestations of HACS. We hope this clarifies our reasoning behind the term HACS.
We added the following to the text “The manifestation of HACS might refer to the possible presence of CS as a neural mechanism in humans. HACS can present itself in many symptoms and / or manifestations such as altered CPM, decreased pain thresholds, increased temporal summation and the presence of widespread pain) but can, until today, not directly be demonstrated in humans[24].” (P31 Line 474-478)
b. What would be the internal consistency of a specific assessment method for HACS? The norm in QST studies is to observe a modality-specific response where different methods for testing may show different responses in the same subjects (Abrishami A. Anesthesiology 2011, Arendt-Nielsen L J Pain 2009, Hübscher M- Pain 2013, Domenech-Garcia V. 2020) e.g., normal PPTs but reduced CPM effect, increased referred pain response to pressure stimulus but normal PPTs, etc. If a term referring to CS manifestations was used, it would be clear that different assessment methods are required for different CS manifestations…
Response 3b: It is difficult to say what the internal consistency is of a specific HACS assessment method, since there is no gold standard for the proxy assessments of CS. Changed provocation findings that may be interpreted as reflecting neural hyperexcitability, may together with other assessment methods pointing towards the presence of possibly central sensitization, contribute to the assumption that HACS may be an underlying factor. Because different expressions of HACS can be identified with different assessment methods we therefore would not suggest to only use one assessment method, but to use a combination of tests. As we say in the grading system: multiple indicator tests should be positive to suggest the presence of HACS: “The second level is based on the indicator tests (see Table 6). With an increasing number of positive indicator tests, as described in this review, the description of “probable” HACS may be relevant. When one or no indicator tests are fulfilled, the patient has “unconfirmed” HACS. Among the described tests to assess HACS in this review, there are assessments that can be used as an indicator test for HACS in patients with CLBP. Candidates are the CSI, PPT, CPM and TS. It should be noted that there is currently only a cut-off value for the CSI established based on previously performed validation studies. For the other indicator tests the validity should be established to be able to distinguish between the presence or absence of HACS. Future studies should also be aimed at assessing the validity of multi-modal test batteries to assess HACS.“ P35, Line 630-639
c. The grading system: Since it treats CS as a diagnose for humans, it is likely that the algorithm/tool remain useless for years. We have no idea when there will be available methods to directly assess CS in humans. Even considering forward-backward translation research between animal studies assessing CS and human studies assessing CS manifestations aiming at validating assessment methods, there are still many assessment methods so that it would take very long time to demonstrate that all methods are real and valid proxies of CS.
Response 3c: Thanks for bringing up this issue. We agree that CS cannot and should not be seen as a diagnosis. However the idea is that the more CS manifestations can be demonstrated, the more susceptible is the presence of HACS. The suggestion of the HACS grading systems is that raising the possibility of HACS to the level of probable, is not in doing one specific assessment, but in following a process using various assessment methods that help building the likely presence of HACS. Standardizing this process may help in gaining probability as a proxy in the absence of a gold standard. Our proposed HACS grading system, that should be elaborated on, maybe useful until a gold standard or reference test is available, or even may be useful if a gold standard is developed but not available for each patient. Therefore we propose to reconsider previously used assessment methods and to reassess the methods that are currently used. These methods can be improved, possibly by combining different assessment methods. We have added your discussion point to the discussion section: “HACS cannot and should not be seen as a ‘diagnosis’ in humans. Assessment of HACS via the grading system will raise the possibility of the presence of HACS which is based on the different manifestations and symptoms related to HACS. In addition, raising the possibility of HACS is not based on a single, specific, assessment but is based on following a process using various assessment methods. Future research is needed to further validate, to improve and expand the initial grading system for HACS. This to create a uniform process in the assessment of HACS. This will lead to a more comparable (clinical) assessment of patients, which will benefit daily clinical practice as well as research on this topic. As an example of such studies, previously performed forward and backwards translations between animal studies and humans studies regarding the assessment of CS showed that the cuff-algometer test for the assessment of CPM is similar between to what is assessed in animals (rats) and in humans[126].” P35, Line 650-661
Minor comments
- Since CS definition is key in this review, I would thank to see some of the CS definitions summarized in table 3 criticized.
For example:
a. “Clark, 2018 [49]”. “Central sensitization involves facilitation of peripheral stimulus processing and alterations in descending inhibitory control of nociceptive input to the brain”. Central sensitization does not necessarily involve altered descending inhibitory control.
b. Also, Clark, 2019 [50] “A dysregulation of the central nervous system causing neuronal hyperexcitability, characterized by generalized hypersensitivity of the somatosensory system to both noxious and non-noxious stimuli”. Central sensitization following injury is first an adaptive mechanism in the central nervous system, not a dysregulation of it.
c. The same with Cuesta-Vargas 2016 [51].
d. Huysmans 2018 [55]: “CS can be defined as a process of abnormal and intense enhancement of pain caused by…” No, CS cannot be defined in such a way. CS is not an enhancement of pain, it may just be related with it. The references included in this one (Staud 2007 [97], Meeus 2008 [98], Meeus 2007 [99]…) have the problem that many other definitions have: They describe CS assuming that CS is based on alterations in all QST modalities, “This central hyperexcitability is associated with altered sensory processing in the brain, malfunctioning of endogenous pain inhibitory system, increased activity of pain facilitatory pathways, temporal summation of second pain and/or wind-up, which leads to dysfunctional endogenous analgesic control” but intra and inter-individual variability in the response of patients with CS manifestations to different QST modalities is high.
e. Kregel 2016 [59]:”CS manifests as...secondary punctuate or pressure hyperalgesia”. This is true, but secondary hyperalgesia would be an adapted (not a dysregulated nor pathophysiological process) response to damage/nociceptive stimulation.
f. Mehta 2017 [63]: No, CS is not “the main cause of chronic pain conditions”.
g. Miki 2020 [65]: ”functional dysregulation of the central nervous system to normal or subthreshold afferent input”. Again, if there is damage, CS is rather a functional regulation. The same with Neblett 2017 [67]. h. Smart 2012 [71]. “Central sensitization pain (CSP) refers to…”. Epistemologically wrong. There is no such a thing as CS pain. Perhaps one critic may address most of these definitions at the same time. The paper would improve if the problematic of these definitions was addressed in the paper.
Response 4: There is no gold standard for central sensitisation and multiple definitions are being used which also can be inadequate or contradictory, as shown in our review and in your comment. Without a gold standard it is very difficult to comment on the definitions used by the different authors, other than there is no reference standard. We are not able to criticise the definitions because there is no scientifically evidenced basis for critics in humans (yet).
This is added to the text: “Across all the studies concerning HACS, a wide variety of definitions for CS are described. At this moment it can’t be said which definition of CS in patients is to be used. Most definitions as described in the included studies in this review are more explanatory rather than really defining central sensitisation with all the associated issues. This because central sensitisation is more than the outcome of just one assessment. Moreover, central sensitisation might also reflect an adaptive primary mechanism or may occur as a secondary response following an (acute) injury[80]. Finally, (nociplastic) pain is not the same as central sensitisation and central sensitisation might not be the main cause of chronic pain[116]” (P30 line 444-452)
- Terms consistency i. Page 24, Line 335: “The widespread pain index (WPI) was used in one study to determine the extent that centralised pain had spread throughout the body as a possible result of HACS [43]”. Due to the concept of “centralised pain” is only used twice in the manuscript, this sentence may be confusing regarding if it refers to pain related to CS or to pain centrally distributed (i.e. pain in the back but not in the extremities)…please, consider changing the concept and be consistent when referring to the type of pain. This comment connects with comment 2.
Response 5: To make it more clearly the three instances in which the term “centralised pain” was used on page 24 the word “centralised” is deleted to make this sentence more clear. This is done except for P2, Line 53, where it is used as a term that was previously been used before the terms nociplastic pain and central sensitisation where introduced.
Discussion
- Page 25, Line 416: “Evidence supporting the preference for a particular assessment method was not found”. How the authors would expect to find a particular method for assessing HACS, if HACS can manifest as many different observations (e.g., reduced CPM, reduced PPTs, increased temporal summation of pain)? Please elaborate on this.
Response 6: Based on this systematic review we thought one or a combination of assessment methods might be used more preferably, but the wide variety of assessment methods that were found did not give a clear preference to any of the assessment methods. The suggested different observations (reduced CPM, reduced PPT, increased TS) are not measurable with one test. Therefore a combination of assessment methods is also a possibility. This is added to the text: “Evidence supporting the preference for a particular assessment method or a combination of assessment methods was not found.” P30, Line 435-436. And also shown in P35, Line 630-632: “With an increasing number of positive indicator tests, as described in this review, the description of “probable” HACS may be relevant.” and P35, Line 638-639: “Future studies should also be aimed to assess the validity of multi-modal test batteries to assess HACS.”
- Substituting in most part of the paper CS by HACS can generate small errors such as in Page 27, line 527: “Both human and animal studies would suggest presence of HACS changes over time depending on pain presence and pain progression”. Human assumed central sensitization couldn´t be in animals. Page 28, Line 548 “Until HACS can be demonstrated in humans…” if it is human assumed central sensitization…the sentence makes no sense. If HACS was demonstrated in humans, it would just be CS. Please reformulate or decide to not substitute CS by HACS systematically in the paper.
Response 7: We changed the wording from “HACS” to “CS” when appropriate.
On the syllogism that a better definition of HACS ïƒ uniform way of assessing HACS ïƒ better treatments…Do the authors think that we can have a uniform CS definition for humans? If we had such a definition, do the authors think that we could use CS as a diagnosis? If we used CS as a diagnosis, would we be able to provide specific treatments for CS? Can the authors elaborate on this?
Response: Thank you for these good discussion points. We think there could be a uniform CS definition for humans. To find a uniform definition for HACS more discussion is needed within the field of HACS. A gold standard would also help to find a uniform definition for HACS, and if a gold standard is available because CS can be demonstrated in humans, the added verbs Human Assumed can be deleted. If CS can be demonstrated, still CS is a neural mechanism that may be related to a pain mechanism, and even than the relationship with pain has to be demonstrated. If CS can be demonstrated to be present and related to pain, we expect that CS can be more specifically targeted for identification and for (development of) specific treatment. As said before (see response 3c).
We would love to discuss this further in a live conversation.
Page 30, Line 600: What´s the point in claiming for normative and cut-off values in markers that are highly variable among the healthy population? Could the authors explain the rationale? This would be very interesting for readers and the pain scientific community.
Response: Thank you for your question. If markers are highly variable in healthy subjects while a marker should help identifying or indicating for a disorder, we should wonder if this marker is an appropriate one. On the other hand, maybe the focus should not be on cut-off scores but a profile/model based on multiple indicator tests.
To clarify this in the writing the sentence has been changed into: “Candidates are the CSI, PPT, CPM and TS. It should be noted that there is currently only a cut-off value for the CSI established based on previously performed validation studies. For the other indicator tests the validity should be established to be able to distinguish between the presence or absence of HACS. Future studies should also be aimed at the validity of multi-modal test batteries to assess HACS.” P35, Line 633-638.

Reviewer 2 Report
It is judged that the definition , the assessment, and the prevalence of central sensitization in patients with chronic low back pain were well organized and analyzed.
As the authors mentioned, it can be said that it is regrettable that this study did not provide a clear definition of HACS and evaluation methods.
However, I think this manuscript is useful in that it can be a basic data for future research on definition and evaluation of HACS.
Author Response
See the attached document

Round 2
Reviewer 1 Report
The authors have adequately adressed all points.